# Multiplexed volumetric CLEM enabled by scFvs provides insights into the cytology of cerebellar cortex

Xiaomeng Han [1] ✉, Xiaotang Lu [1] ✉, Peter H. Li [2], Shuohong Wang[1], Richard Schalek[1], Yaron Meirovitch[1], Zudi Lin [3], Jason Adhinarta [4], Karl D. Murray[5], Leah M. MacNiven[5], Daniel R. Berger [1], Yuelong Wu [1], Tao Fang [6], Elif Sevde Meral [7], Shadnan Asraf[8], Hidde Ploegh [6], Hanspeter Pfister[5], Donglai Wei[4], Viren Jain [2], James S. Trimmer [5] & Jeff W. Lichtman [1] ✉

Mapping neuronal networks is a central focus in neuroscience. While volume electron microscopy (vEM) can reveal the fine structure of neuronal networks (connectomics), it does not provide molecular information to identify cell types or functions. We developed an approach that uses fluorescent single-chain variable fragments (scFvs) to perform multiplexed detergent-free immunolabeling and volumetric-correlated-light-and-electron-microscopy on the same sample. We generated eight fluorescent scFvs targeting brain markers. Six fluorescent probes were imaged in the cerebellum of a female mouse, using confocal microscopy with spectral unmixing, followed by vEM of the same sample. The results provide excellent ultrastructure superimposed with multiple fluorescence channels. Using this approach, we documented a poorly described cell type, two types of mossy fiber terminals, and the subcellular localization of one type of ion channel. Because scFvs can be derived from existing monoclonal antibodies, hundreds of such probes can be generated to enable molecular overlays for connectomic studies.

Mapping neuronal networks that underlie behavior is a central focus in neuroscience. Techniques that reveal neurons' structure and molecular components and their connections have expanded over the last several decades. Automated high-resolution volume electron microscopy (vEM)[1] provides volumetric neuronal ultrastructure but omits important contexts related to molecular diversity. For example, vEM lacks molecular markers for cell or synapse types or the localization of specific molecules related to neuronal physiology.

One approach for obtaining structural and molecular information in the same sample is immunolabeling. Although it is possible to label specific molecules directly in electron microscopy images[2-4], these immunolabeling strategies have several technical challenges. Pre-embedding immunolabeling[5] requires membrane permeabilization for antibody access that perforates cell membranes, causing them to be discontinuous in EM images, which is incompatible with neural circuit tracing. Post-embedding methods (typically after sectioning) suffer from both the loss of antibody binding due to denatured antigen epitopes and the challenges of using aqueous immunoreagents with hydrophobic resin-embedded sections[6]. Alternatively, molecular labeling via transgenically engineered, EM-visible molecular tags like

[1]Department of Molecular and Cellular Biology, Harvard University, Cambridge, MA, USA. [2]Google Research, Mountain View, CA, USA. [3]School of Engineering and Applied Sciences, Harvard University, Cambridge, MA, USA. [4]Computer Science Department, Boston College, Chestnut Hill, MA, USA. [5]Department of Physiology and Membrane Biology, University of California Davis School of Medicine, Davis, CA, USA. [6]Program of Cellular and Molecular Medicine, Boston Children's Hospital, Boston, MA, USA. [7]Bezmialem Vakif University School of Medicine, Istanbul, Turkey. [8]School of Public Health, University of Massachusetts Amherst, Amherst, MA, USA. ✉e-mail: xiaomenghan@fas.harvard.edu; xiaotang_lu@fas.harvard.edu; jeff@mcb.harvard.edu

miniSOG[7], APEX[8–10], and EMcapsulins[11] can be used to label cell types in vEM samples. This genetically encoded approach is more challenging as a means of localizing a protein of interest. Immunolabeling remains the most straightforward way to discover where a protein is located in a cell.

In many cases, immunolabeling can also identify cell types, assuming that the protein of interest is expressed in the cell soma. Immunofluorescence and vEM, when performed sequentially on the same sample (volumetric correlated light and electron microscopy, vCLEM) is a useful way to localize proteins and cell types within the context of ultrastructure. Because multiplex labeling (e.g., with many colors) can be easily achieved by fluorescence LM using spectral unmixing[12,13], vCLEM can potentially incorporate more molecular and functional information into vEM datasets than directly tagging within electron microscopy images.

One challenge of vCLEM is the requirement of permeabilizing detergents for the antibodies to gain access to intracellular sites. Detergents can compromise membrane structure. Attempts to circumvent the need for permeabilization include the post-resin labeling techniques used in array tomography[14]. This CLEM approach is effective, but the modified heavy metal staining compromises the ultrastructure, and the water-permeant resin is too soft for the ultrathin (30–40 nm) sectioning needed for connectomic studies. Detergent-based permeabilization is also avoided by the use of nanobodies as immunolabels[15], which owing to their small size, diffuse into tissue samples without permeabilizing agents like Triton X-100. Nanobodies are still limited and difficult to generate for certain brain markers. Extracellular space (ECS) preservation also facilitates the diffusion of standard mammalian antibodies (Immunoglobulin G, IgGs) into brain cells without detergent treatment[16,17]. However, fluorescent labeling with IgGs using a secondary antibody limits the number of color probes in the same sample. Alternatively, the full-length antibody can be tagged directly with fluorophores covalently linked to a particular amino acid moiety. However, it is possible that the large size of IgGs will limit their diffusibility into volumetric CLEM samples suitable for connectomics.

The goal of this study was to evaluate another type of small fluorescently conjugated antibody derivatives (single-chain variable fragments (scFvs)) as a method to label intracellular epitopes in large-volume samples without the need for detergents so that volumetric CLEM could be accomplished. We were interested in a method that allowed multiple scFvs to be used in the same sample to have multiple molecular labels localized within the same EM volume. In this study we showed that the method was successful, and the volumetric fluorescent and electron microscopy image data were of high quality. This scFv immunolabeling method may hold promise for routine linking molecular information to connectomic information obtained from the same tissue samples.

## Results

### Characterization of detergent-free scFv immunolabeling

Our first goal was to generate scFvs from full-size IgG monoclonal antibodies (mAbs) that retain the binding specificity of the parental mAb. ScFvs are built by recombinantly linking the $V_H$ and $V_L$ domains of sequenced mAbs[18,19] via a flexible peptide linker[20] (Fig. 1a). When the $V_H$ and $V_L$ fold and assemble correctly (~60% of the time), they retain their ability to bind to the antigen[18]. Because scFvs are 1/5 the size of conventional antibodies, we evaluated if they could diffuse into tissue samples without detergent-based permeabilization, as is the case for nanobodies. Unlike nanobodies, however, there are extensive collections of well-characterized mAbs that label neuronal cell types or signaling molecules[21], enabling the development of an extensive collection of scFvs that can be used to label nervous system samples. Moreover, because scFvs are produced recombinantly, they can be engineered so that different fluorescent dyes can easily be conjugated for multiplex imaging.

We first generated an anti-green fluorescent protein (GFP) scFv based on the sequence of the anti-GFP mouse mAb N86/38[21]. This mAb binds to both GFP and the GFP derivative YFP. The sequence was used to construct an scFv in which the $V_H$ and the $V_L$ domains were connected via a 3× flexible linker (Fig. 1a). A sortase tag was added for dye conjugation (see "Methods"). This scFv was conjugated directly with the red fluorescent dye 5-TAMRA (Fig. 1a). We tested the anti-GFP scFv with our detergent-free immunofluorescence protocol (see "Methods") on the cerebral cortex from YFP-H mice[22] (Fig. 1b and Supplementary Fig. 1a). We found that the scFv probe retained the binding properties of the parental mAb and penetrated aldehyde-fixed brain tissue without the need for detergent treatment. Only a few thinner neuronal processes, possibly myelinated axons, were not well labeled with the scFv (arrows in Fig. 1b and Supplementary Fig. 3). We compared the tissue penetration depth of this scFv, an anti-GFP polyclonal antibody, and an anti-GFP nanobody (Supplementary Fig. 1c, d). ECS preservation increased the penetration depth of this scFv from 60 μm to >100 μm, nearly the penetration depth of nanobodies and far greater than the penetration of full-length polyclonal IgG antibodies.

We then produced fluorescent scFvs for the endogenously expressed proteins calbindin (CB), postsynaptic density protein 95 (PSD-95), neuropeptide Y (NPY), vesicular glutamate transporter 1 (VGluT1), glial fibrillary acidic protein (GFAP), Potassium voltage-gated channel subfamily A member two ($K_v$1.2), and parvalbumin (PV) based on seven well-characterized mouse mAbs[21,23] (see Table 1). Because previous studies showed that an scFv's performance improves as the length of the linker increases[24], we chose a 4× linker rather than a 3× linker for 6 of the 7 scFvs (see Table 1—linker length). All these probes labeled their targets in detergent-free immunolabeling with the standard fixation method without ECS preservation (Figs. 1c–e and 2a and Supplementary Fig. 2a). Except for the anti-GFAP scFv, all scFvs were validated in a previous study[25]. We performed additional validation of these probes by comparing them to their respective parental mAbs or commercial polyclonal antibodies (pAbs) in immunofluorescence (Supplementary Figs. 9–15). All probes generated detergent-free immunofluorescence patterns similar to their parental mAbs or pAbs (Supplementary Figs. 9–15). In the case of CB and PV, the scFvs produced more consistent labeling results for the cell bodies and nuclei of Purkinje cells than labeling with full-size mAbs, which cannot label the nuclei of Purkinje cells except with prolonged incubation or special tissue cutting orientation to better expose the nuclei (Supplementary Fig. 15c, f, l, n).

CLEM is frequently accomplished using the mild detergent saponin[26,27], raising the possibility that full-length IgG antibodies can be used as an alternative to scFvs. We, therefore, repeated the anti-GFP scFv penetration test (see "Methods") for the anti-CB and anti-PV scFvs on a 1-mm thick cortical section in the absence of detergent. We compared the results with the labeling of the dye-conjugated parental mAbs in the presence of 0.05% saponin. Saponin at 0.05% concentration allowed the mAbs to penetrate 250 μm deep after a 1-week incubation with the labeled mAb (Fig. 1g and Supplementary Fig. 4). In contrast, the anti-PV and anti-CB scFvs penetrated throughout the entire block (i.e., at least 500 μm) in the absence of detergent (Fig. 1f and Supplementary Fig. 4). A comparison of the ultrastructure of samples without and with saponin detergent treatment (7 days) showed substantial differences. The ultrastructure of the sample without detergent was well-preserved (Fig. 1h). Saponin for seven days, in contrast, generated poorer ultrastructure with membrane breaks and abnormal appearing vesicle-filled axonal terminals (Fig. 1i).

A higher concentration of saponin (0.1% and 0.2%), but combined with a shorter (1-day) incubation, with the aim of preserving ultrastructure, did not provide as good penetration of IgGs as that obtained for scFvs in the absence of saponin treatment, and the ultrastructure was still compromised (Supplementary Fig. 5). The only detergent concentration that preserved ultrastructure was 0.05% saponin but

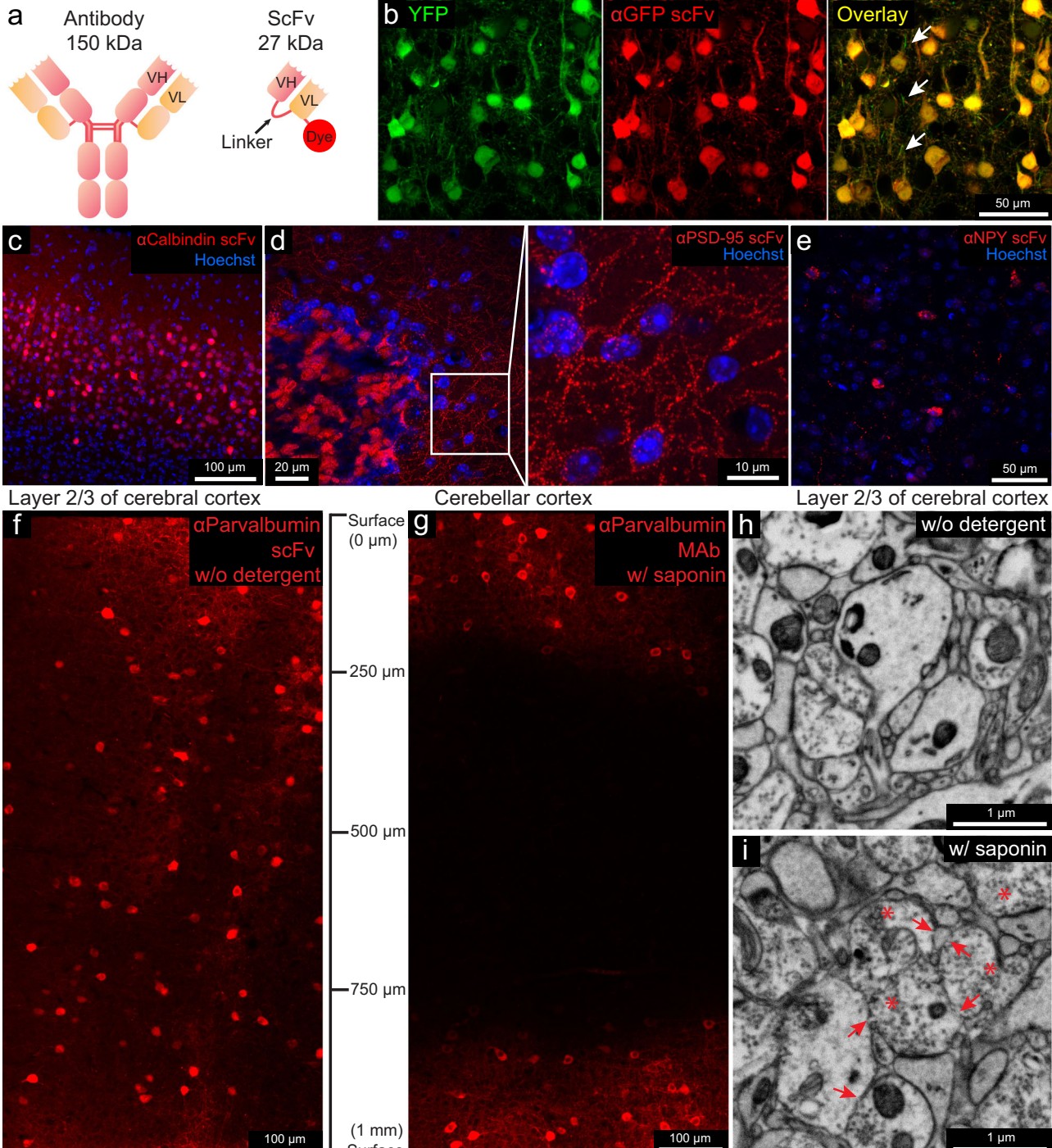

**Fig. 1 | Fluorescent scFv probes label brain tissues without detergents, preserving electron microscopy ultrastructure. a** Schematic of a full-length IgG antibody and an scFv probe with a fluorescent dye. **b** Confocal images from the cerebral cortex of a YFP-H mouse labeled with a GFP-specific scFv probe conjugated with 5-TAMRA (*n* = 3 experiments, all experiments mentioned refer to independent experiments). Arrows indicate unlabeled thinner neuronal processes. **c** Layer 2/3 of the cerebral cortex labeled with a calbindin-specific scFv probe (*n* = 3 experiments).

**d** Cerebellar cortex labeled with PSD-95-specific scFv (*n* = 3 experiments), with an enlarged inset. **e** Cerebral cortex labeled with an NPY-specific scFv (*n* = 3 experiments). **f**, **g** Penetration depth comparison of a parvalbumin-specific scFv without detergent versus parental mAbs with 0.05% saponin (*n* = 2 experiments). **h**, **i** Ultrastructure comparison after 7 days incubation without detergent versus 0.05% saponin (*n* = 2 experiments). Arrows indicate membrane breaks; asterisks indicate abnormal vesicle-filled axonal terminals.

with a 1-day incubation the mAb would only penetrate 30 μm, compared to scFvs without detergent reaching the middle of the 300-μm section while preserving excellent ultrastructure (Supplementary Figs. 5a and 6).

The better penetration ability of scFvs compared to mAbs (even in the presence of detergents) may be explained by their greater ability to cross cell membranes that are fixed with formaldehyde fresh prepared from paraformaldehyde and glutaraldehyde. These fixatives are known to perturb the integrity of cell membranes[28,29], which may cause small gaps to allow scFv to pass through. To explore this mode of entry further, we immunolabeled transfected HEK293T or COS-1 cells fixed with formaldehyde only or together with glutaraldehyde. ScFvs

**Table 1 | Fluorescent scFv immuno-probes generated in this work**

| Target | mAb clone no. | Linker length | Fluorescent dyes conjugated | Link of expression plasmid at Addgene |
|---|---|---|---|---|
| GFP | N86/38 | 3× | 5-TAMRA | https://www.addgene.org/204419/ |
| Calbindin | L109/57 | 4× | 5-TAMRA, Alexa Fluor 488, Alexa 594 | https://www.addgene.org/204421/ |
| GFAP | N206B/9 | 3× | 5-TAMRA, Alexa 488 | https://www.addgene.org/204420/ |
| VGluT1 | N28/9 | 4× | 5-TAMRA, Alexa Fluor 532 | https://www.addgene.org/204424/ |
| PSD-95 | K28/43 | 4× | 5-TAMRA | https://www.addgene.org/204422/ |
| $K_v1.2$ | K14/16 | 4× | Alexa Fluor 594 | https://www.addgene.org/204425/ |
| Parvalbumin | L114/81 | 4× | 5-TAMRA, Alexa Fluor 647 | https://www.addgene.org/204423/ |
| Neuropeptide Y | L115/13 | 4× | Alexa Fluor 594 | https://www.addgene.org/204426/ |

penetrated these cells and strongly labeled intracellular targets in the absence of detergent (Supplementary Figs. 7a, b and 8). Interestingly, scFvs also penetrated live cells without chemical fixation (Supplementary Fig. 7c).

## Multiplexed vCLEM based on scFv-enabled immunofluorescence using linear unmixing

To attempt multiplexed vCLEM, we chose five fluorescent probes with different targets and linked to different fluorophores (anti-CB-Alexa Fluor 488, anti-VGluT1-Alexa Fluor 532, anti-GFAP-5-TAMRA, anti-$K_v$1.2-Alexa Fluor 594, and anti-PV-Alexa Fluor 647) to simultaneously label a 120-μm-thick cerebellar sample from Crus 1, a rarely studied region in the cerebellum. Because these fluorophores have overlapping excitation and emission spectra (Supplementary Fig. 16a), we used spectral unmixing on a laser scanning confocal microscope for multiplex imaging. A lambda stack of the multicolor sample with a depth of 52 μm was imaged by confocal microscopy (Zeiss LSM 880) using a spectral detector (Fig. 2b and Supplementary Fig. 16b). Because a fluorophore's reference spectrum (required for linear unmixing) may vary depending on brain regions[13] we acquired a lambda stack for each fluorophore in individually labeled samples from the same region as the multi-labeled sample (Fig. 2b and Supplementary Fig. 16c). Using a linear unmixing algorithm and the reference spectra of all five dyes (Fig. 2b and Supplementary Fig. 16c), the lambda stack of the multi-labeled sample was separated into five fluorescent probe channels (Fig. 2b and Supplementary Fig. 17a). In some cases, spectral unmixing can be accomplished more efficiently by extracting reference spectra from the multi-labeled sample[30]. Still, these approaches were unsatisfactory with this sample (Supplementary Fig. 17b, c).

The spectrally unmixed labeling pattern for each probe was distinct and resembled what we found in the individually labeled samples (Fig. 2c and Supplementary Figs. 17a and 18). In our sample, the signal for cell nuclei stained with Hoechst dye was acquired without spectral unmixing using 405-nm laser excitation. The short excitation and emission wavelength of Hoechst scattered strongly and lost intensity dramatically with depth requiring a tenfold increase in laser power for the deepest parts of the volume. However, we found in a separate identically prepared sample, Hoechst could be combined with the other linear unmixing channels with equally good results (Supplementary Fig. 19). Overall, we used linear unmixing to acquire a multi-color fluorescence image volume (304 μm × 304 μm × 52 μm), showing the labeling of the five scFv probes (Fig. 2c, d and Supplementary Fig. 20). Each fluorescence channel is highly specific for the labeling pattern of the corresponding scFv. For example, the labeling of Bergmann fibers (arrow) by the anti-GFAP scFv and the labeling of the axons in pinceau structures (arrowhead) by the anti-$K_v$1.2 scFv are easily distinguishable (Fig. 2d). After confocal imaging was completed, the sample was stained with heavy metals and embedded in resin (see "Methods") and imaged in X-ray μCT with an isotropic resolution (voxel size 1.15 μm³) (see "Methods").

The sample was then cut into ~4000 serial 30 nm ultrathin sections using an automated tape-collecting ultramicrotome[31].

A low-resolution overview image subvolume of 1 mm × 1 mm × 60 μm with a voxel size of 150 nm × 150 nm × 1.35 μm, was obtained with a single-beam scanning electron microscope from the half of all the serial sections where confocal imaging was of high quality. This was followed by multi-beam scanning electron microscope imaging to collect a high-resolution volume of 164 μm × 200 μm × 26 μm at a resolution of 4 nm × 4 nm × 30 nm (Fig. 3a), comprising the superficial 22% of the serial sections. Tissue ultrastructure appeared well-preserved throughout the 60-μm subvolume (60-μm is the middle of the section) that included all layers of the cerebellar cortex except for some mild artifacts in the molecular layer (Fig. 3a, 1–4 and Supplementary Fig. 21).

Co-registration between the stacks of fluorescence images and EM volumes was accomplished by visually finding common fiduciaries (such as blood vessels) in the μCT volume and confocal volume. This co-registration provided an intermediate step to guide the trimming of the block to the relevant region prior to ultrathin sectioning (see Methods). To target the location for high-resolution EM imaging, the confocal stack to the low-resolution EM stack were registered. This morphing registration overcame the rotation, tilt, stretch, and warping that distort the EM sections[32]. For the final, high-precision co-registration between the six-color fluorescence confocal volume and the high-resolution EM volume, 188 manually picked landmark points that corresponded to the same sites in the two volumes were placed on blood vessels, cell nuclei, cell bodies, and axons (Supplementary Fig. 22). A 3D transformation of the fluorescence volume was produced by a thin plate spline interpolation based on the point correspondences (BigWarp in FIJI)[33]. The transformed fluorescence volume was then overlaid with the high-resolution EM volume to create the vCLEM dataset (Fig. 3a). The fluorescence signals from the molecular markers corresponded with objects visible in the EM ultrastructure throughout the volume (Fig. 3b and Supplementary Fig. 23). Figure 3b shows an example slice in which CB (green) correlates with Purkinje cells and their dendrites, GFAP (red) with astrocytes around blood vessels, VGluT1 (cyan) with parallel fiber boutons in the molecular layer as well as mossy fiber terminals in the granule layer, $K_v$1.2 (yellow) with the pinceau surrounding Purkinje cell axons, and PV (magenta) with both Purkinje cells and molecular layer interneurons.

In order to facilitate visualization of the vCLEM dataset, we imported it into Neuroglancer[34] and generated the link (provided in the source data) where each fluorescence channel and the EM channel can be visualized separately or simultaneously, and navigation through slices and different resolution levels can be carried out easily.

## 3D reconstruction of cells and subcellular structures identified by scFv labeling in the cerebellar cortex

An important question is whether the EM data, with six-color immunolabeling superimposed, is of sufficient quality to be successfully segmented by automatic means. Two different methods of automatic segmentation were successful. We used a flood-filling network[35] pretrained on a different dataset[36] at either 32 nm or 16 nm and without any ground truth (Supplementary Fig. 24a, b). We also used another

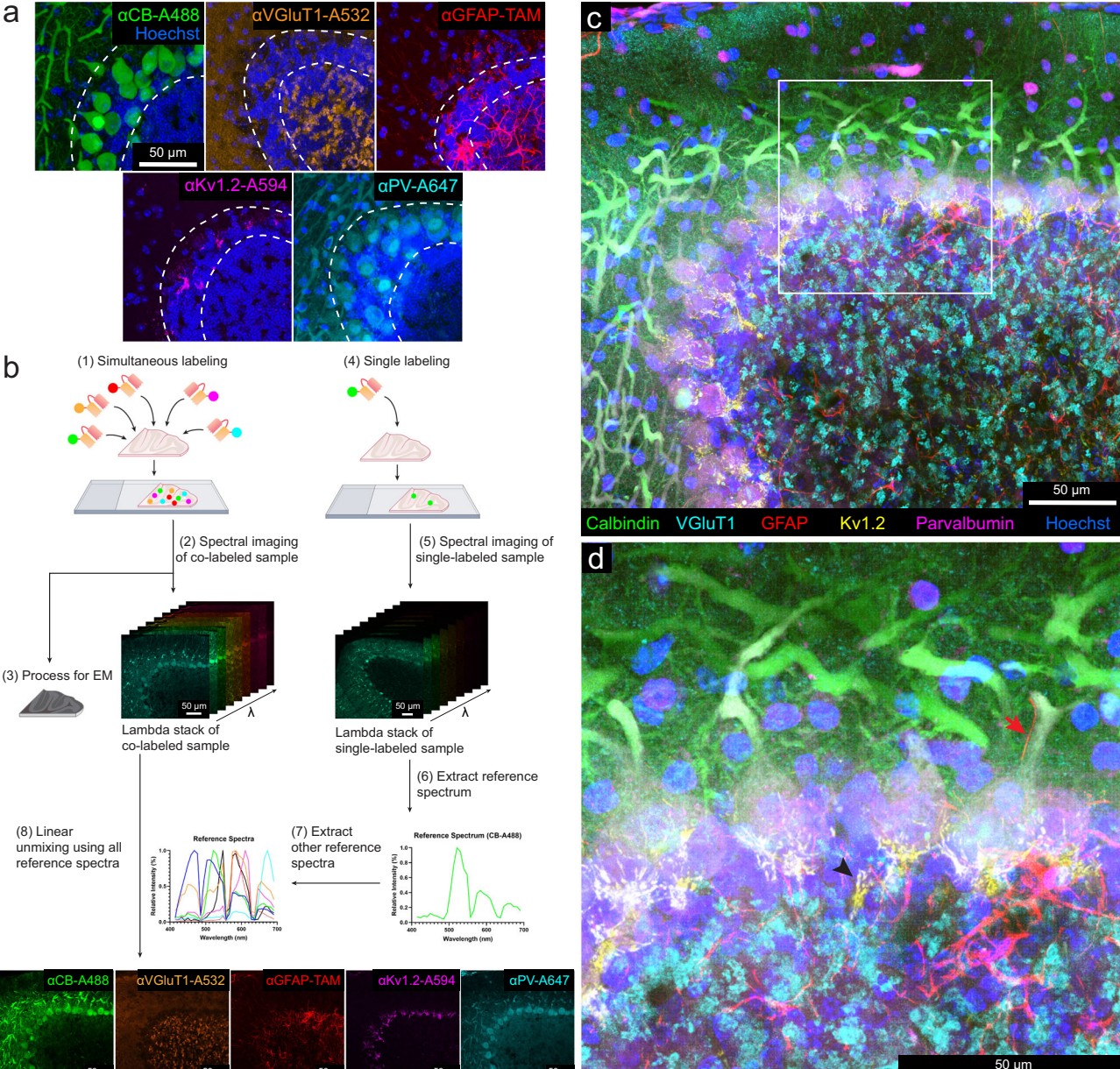

**Fig. 2 | Multicolor immunofluorescence enabled by scFv probes and linear unmixing of confocal micrographs. a** Representative confocal images (*n* = 3 experiments in each category) of different sections from the cerebellum labeled with: a calbindin-specific scFv probe conjugated with Alexa Fluor 488, a VGluT1-specific scFv probe conjugated with Alexa Fluor 532, a GFAP-specific scFv probe conjugated with 5-TAMRA, a K$_v$1.2-specific scFv probe conjugated with Alexa Fluor 594, and a parvalbumin-specific scFv probe conjugated with Alexa Fluor 647. The double dotted lines delineate the Purkinje cell layer. (see Supplementary Figs. 12 and 15 for larger fields of view). CB calbindin, VGluT1 vesicular glutamate transporter 1, GFAP glial fibrillary acidic protein, K$_v$1.2 potassium voltage-gated channel subfamily A member 2, PV parvalbumin, TAM 5-TAMRA. **b** Workflow of multicolor imaging enabled by scFv probes and linear unmixing (see the text). **c** Representative maximum intensity projection of the multicolor fluorescence image stack acquired by linear unmixing of confocal images (*n* = 3 experiments). The signal of each fluorescent dye was pseudo-colored for better visualization. **d** Enlarged boxed inset from (**c**). The arrow indicates a Bergmann fiber (GFAP-positive) adjacent to the main dendrite of a Purkinje cell. Arrowhead indicates sites where axons form a pinceau structure labeled by the K$_v$1.2-specific scFv probe.

method developed in our lab[37,38] to generate high-quality 2D membrane predictions, as well as 2D segmentation at 8 nm resolution (Supplementary Fig. 24c, d) following several rounds of training with manually generated ground truth. The success of the automatic segmentation confirms that the ultrastructure is not compromised by the detergent-free scFv immunofluorescence labeling protocol.

We then analyzed the CLEM data by reconstructing the EM ultrastructure at sites labeled with each of the five molecular markers (CB, GFAP, PV, K$_v$1.2, and VGluT1). All reconstructions were performed on 848 slices of EM images in the vCLEM dataset.

Calbindin (CB): In the cerebellar cortex, CB is expressed by Purkinje cells and some Golgi cells in the granule layer[39–41]. In the vCLEM dataset, labeling of the anti-CB scFv (green fluorescence signal in Fig. 4a and Supplementary Fig. 25) corresponded to Purkinje cell bodies (Fig. 4a, b), a few likely large Golgi cells in the upper granule cell layer[42] (Supplementary Fig. 24b, c), and heavily myelinated axons that presumably arose from Purkinje cells (Supplementary Fig. 25a, inset 2). 3D reconstructions of one of the labeled Purkinje cells (Fig. 4c and Supplementary Fig. 25a) and one of the labeled Golgi cells (Supplementary Fig. 25d) showed that their morphologies were similar to

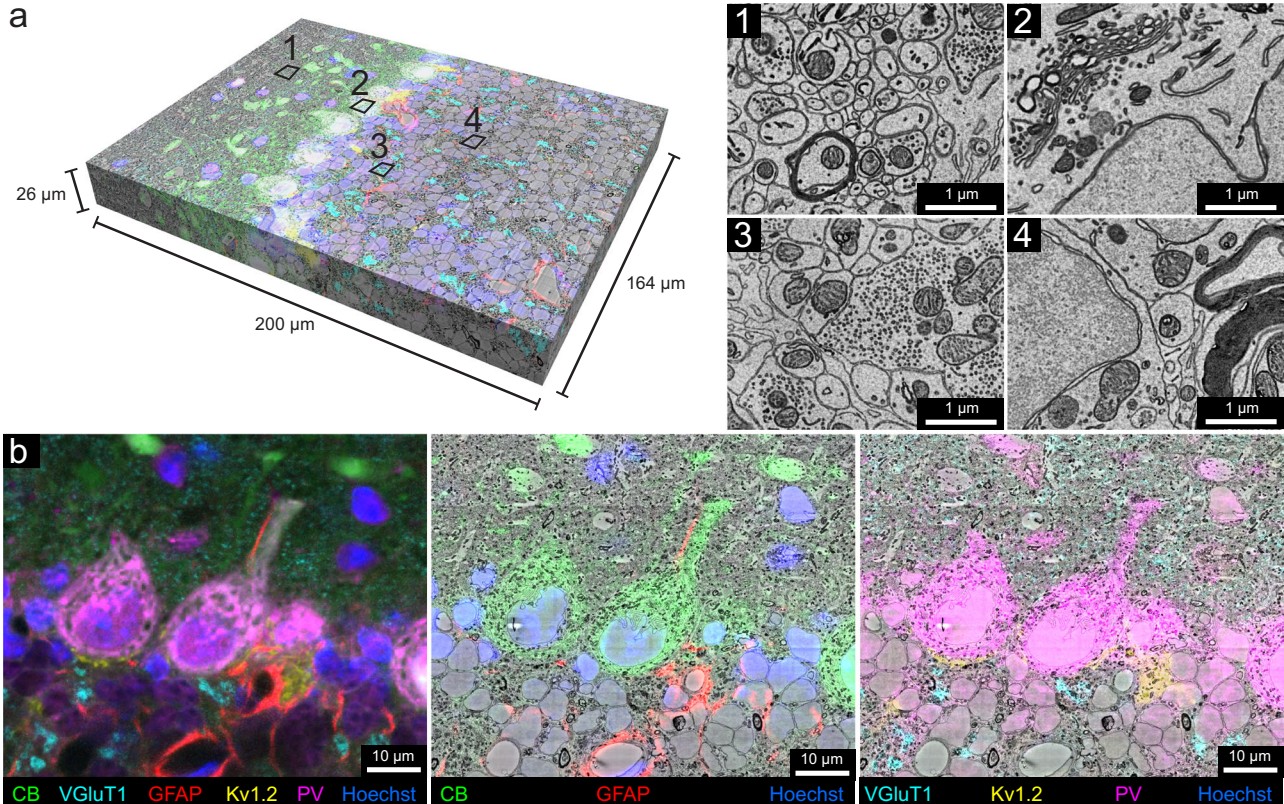

**Fig. 3 | Multicolor volumetric CLEM enabled by scFv-assisted immuno-fluorescence. a** The high-resolution EM volume acquired from the cerebellar lobule, Crus 1 with multicolor immunofluorescence from scFv probes separated by linear unmixing (*n* = 1 experiment). The multicolor fluorescence data was co-registered with the high-resolution EM data. The Neuroglancer link to access the dataset is provided in the source data file. Numbers 1–4 indicate approximate regions where the ultrastructure was examined at high resolution (*n* = 12 experiments). Owing to the absence of detergent in immunofluorescence labeling, fine ultrastructure was preserved throughout the EM volume, such as in the molecular layer (1), in the Purkinje cell layer (2), in the glomeruli in the granule cell layer (3), and in the granule cell bodies (4). **b** Demonstration of the overlay between fluorescence signals and EM ultrastructure. Left panel shows the multicolor six-channel fluorescent image of slice 250 (*n* = 848 slices) of the spatially transformed fluorescence image volume. The middle panel shows three fluorescence channels corresponding to the labeling of CB, GFAP, and Hoechst overlaid onto the EM micrograph of slice 250. Right panel shows four fluorescent channels corresponding to the labeling of VGluT1, K$_v$1.2, PV, and Hoechst overlaid onto the EM micrograph of slice 250. Other examples of fluorescence overlay are shown in Supplementary Fig. 23.

previous Golgi/ultrastructural studies of rat cerebellum[42]. The synaptic inputs from parallel fibers onto Purkinje cells are also obvious (Fig. 4c, d). In addition, we found that the dendritic trees of Purkinje cells in this region (Crus 1) were not perpendicular to the rostral-caudal axis but intersected at the axis at an angle of around 55° (Supplementary Fig. 25a, inset 1).

GFAP: This protein is expressed by astrocytes and Bergmann glial cells whose cell bodies reside in the Purkinje layer and whose processes (also known as Bergmann fibers) extend into the molecular layer[43,44]. As expected, labeling seen with the anti-GFAP scFv (red fluorescence signal in Fig. 4e, i) corresponded to the cell bodies and processes of granule layer astrocytes (Fig. 4e, f) and the vertical Bergmann fibers in the molecular layer (Fig. 4i, j). Although the cell bodies of granule layer astrocytes have a similar appearance to those of granule cells in EM micrographs (Fig. 4f and Supplementary Fig. 26a, insets 1, 2), 3D reconstruction of one of the labeled cells shows the typical morphology of a velate astrocyte[42] that is different from the structure of nearby granule cells, despite their similar cell body ultrastructure (Fig. 4g and Supplementary Fig. 26a). The reconstructed velate astrocyte extended a thin, veil-like glial process between two granule cells (Fig. 4g, h). GFAP labeling was also seen in the processes of Bergmann glia but not in their cell bodies (Fig. 4i and Supplementary Fig. 26b, insets). Two Bergmann glia cell bodies were found by tracing back from their labeled processes (Fig. 4i and Supplementary Fig. 26b).

These cell bodies were adjacent and between two Purkinje cells (Fig. 4k and Supplementary Fig. 26b). Although they were similar in size to nearby basket cells, they were distinguishable by virtue of the lack of infoldings in their nuclear membranes (Fig. 4l).

Parvalbumin (PV): This protein is expressed by molecular layer interneurons (MLIs), Purkinje cells and Purkinje layer interneurons[39,45]. The labeling we observed (magenta fluorescence signal in Fig. 5a and Supplementary Fig. 27a, b) in the molecular layer co-registered with MLIs (Fig. 5b and Supplementary Fig. 27c, d) that have in-folded nuclear membranes. Among PV-positive MLIs (*n* = 22 samples) that we partially reconstructed, we chose three for full reconstruction. These neurons were located at three different levels of the molecular layer (Fig. 5c) and exhibited different morphologies as has been previously described[45].

The PV-positive labeling allowed the examination of a set of neurons in the molecular layer that are PV-negative (Fig. 5f, g and Supplementary Fig. 27f, g). All of these neurons (*n* = 7 samples) showed morphological features consistent with them being a rarely studied type of neuron, known as molecular layer granule cells (MGCs)[42,46,47]. MGCs are sometimes regarded as ectopic granule cells that did not migrate to the granule cell layer during development[42]. A recent study[48] showed that they are in fact relatively common and that they behave electrophysiologically like typical granule cells located in the granule cell layer. The MCG/MLI ratio of our dataset was 32%, which is

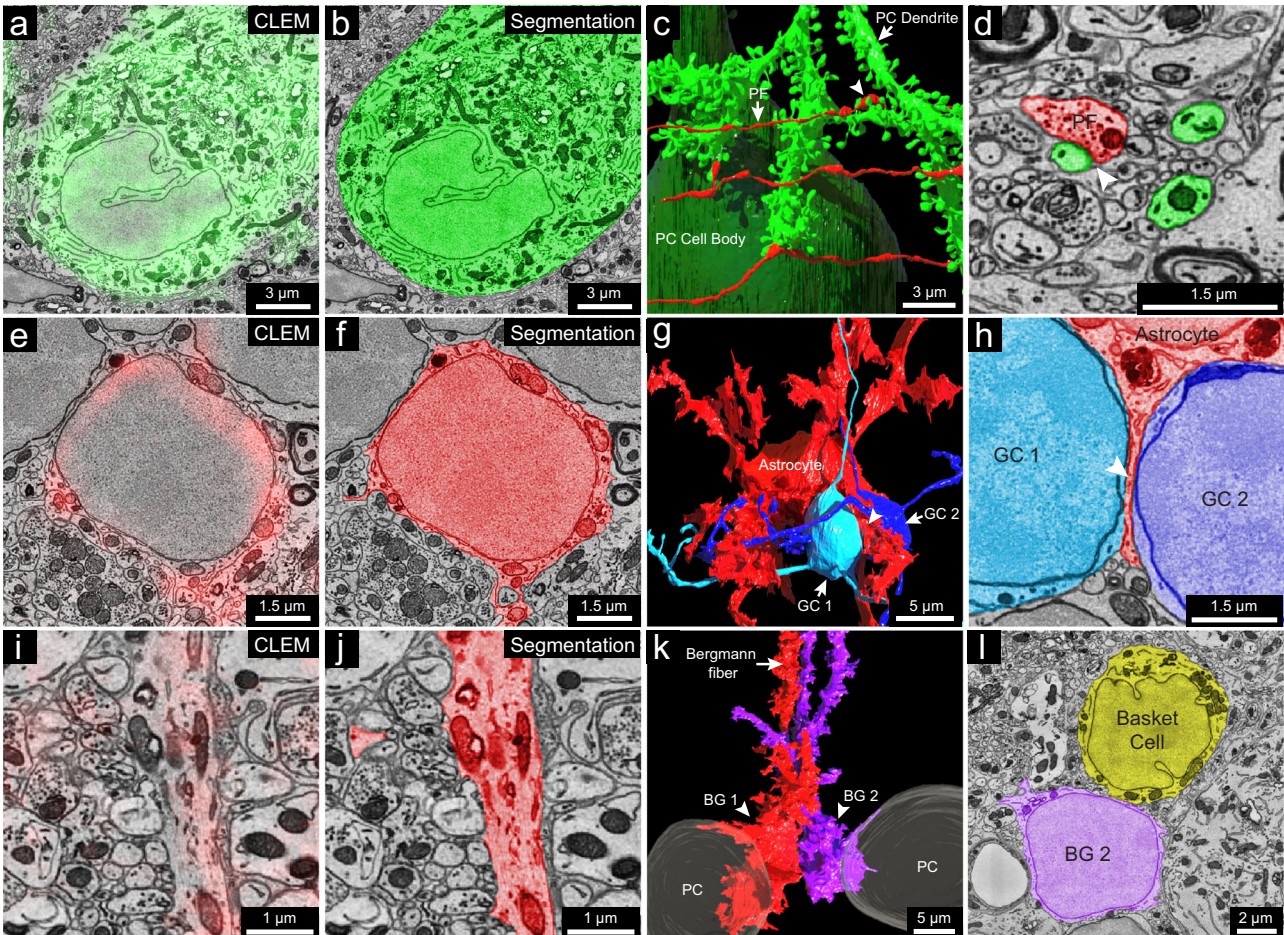

**Fig. 4 | 3D reconstruction of cells labeled by calbindin-specific and GFAP-specific scFv probes. a** 2D CLEM image showing the fluorescence signal (green) of the calbindin-specific scFv probe overlapping with the cell body of a Purkinje cell. **b** EM image showing 2D segmentation (green) of the calbindin-positive Purkinje cell ($n = 1$). **c** 3D reconstruction of the Purkinje cell labeled in a ($n = 1$), with the cell body in dark green and a dendritic branch in light green; three parallel fibers (red) make synapses on three spine heads of the dendritic branch (arrow indicates a parallel fiber (PF); arrowhead indicates a synapse). **d** EM image showing 2D segmentation of the synapse (arrowhead) between a parallel fiber (red) and a spine head of the dendritic branch (green) ($n = 1$). **e** 2D CLEM image showing fluorescence signals (red) of the GFAP-specific scFv probe overlapping with the cell body of a velate astrocyte in the granule cell layer ($n = 1$). **f** EM image showing 2D segmentation (red) of the velate astrocyte in (**e**) ($n = 1$). **g** 3D reconstruction of the velate astrocyte (red) labeled in (**e**) and two nearby granule cells (GC1 and GC2, light and dark blue) ($n = 2$); the astrocyte extends a veil-like glial process (arrowhead) between the two granule cells. **h** EM image showing 2D segmentation of the glial process (arrowhead) between GC1 and GC2 ($n = 1$). **i** 2D CLEM image showing fluorescence signals (red) of the GFAP-specific scFv probe overlapping with a Bergmann fiber ($n = 1$). **j** EM image showing 2D segmentation (red) of the Bergmann fiber in (**i**) ($n = 1$). **k** 3D reconstruction of two Bergmann glial cells (BG1 and BG2) ($n = 2$) traced from their Bergmann fibers labeled by the GFAP-specific scFv probe. **l** EM image showing 2D segmentation of the cell body of BG2 ($n = 1$) and a nearby basket cell ($n = 1$), noting the lack of infoldings in BG2's nuclear membrane compared to the basket cell. $n$ indicates an example.

similar to a previous quantification for the posterior lobe of the cerebellum from which our sample was derived[48]. As mentioned, these neurons lack PV labeling in their cell bodies (Fig. 5f and Supplementary Fig. 27f, g). Their cell bodies share ultrastructural features with typical granule cells. For example, they had round nuclear membranes without infoldings and only a small rim of cytoplasm surrounding the nucleus (Fig. 5g and Supplementary Fig. 27h, i). Reconstruction of three MGCs showed that their 3D morphology was also similar to that of granule cells (Fig. 5h). They possessed several short dendrites, each forming a loosely organized glomerulus (Supplementary Fig. 27j, insets 1, 2). In some cases, the glomeruli were associated with mossy fiber collaterals that reached the Purkinje cell layer or slightly above (Supplementary Fig. 27j, insets 3, 4). This is consistent with previous descriptions of mossy fibers[42]. Surprisingly, the MGC dendrites were also innervated by parallel fibers (Fig. 5h–j). This has not been reported previously and is never the case for typical granule cells.

K$_v$1.2: This protein is a member of the voltage-gated potassium channel subfamily A[49], and shows high expression in the axonal

terminals of the MLIs that form pinceau structures around Purkinje cell axon initial segments (AIS), as indicated by the strong yellow fluorescence signal in Fig. 6a and the reconstruction shown in Fig. 6b, c[50,51]. These axons are of different caliber (Supplementary Fig. 28a-1–7), and none are downstream branches of the same axon, suggesting that they each come from a different MLI cell. K$_v$1.2 also has been reported to be present in small puncta in the granule layer at juxtaparanodes of axons of unknown identity[52]. We examined 22 sites of juxtaparanodal puncta in 22 axons in the granule cell layer. Consistent with previous reports[49,52], we found 15 pairs of puncta that corresponded to the juxtaparanodal portion of nodes of Ranvier (Fig. 6d–j). In addition, we found two sites with three puncta that corresponded to the juxtaparanodal portion of nodes of Ranvier that are at branch points (Supplementary Fig. 28b); Finally, we found five sites with a single punctum, which corresponded to the final myelinated site of an axon where myelination ended, termed a hemi-node of Ranvier (Supplementary Fig. 28c). From the EM volume, we reconstructed these labeled axons. Six out of 22 had *en passant* or

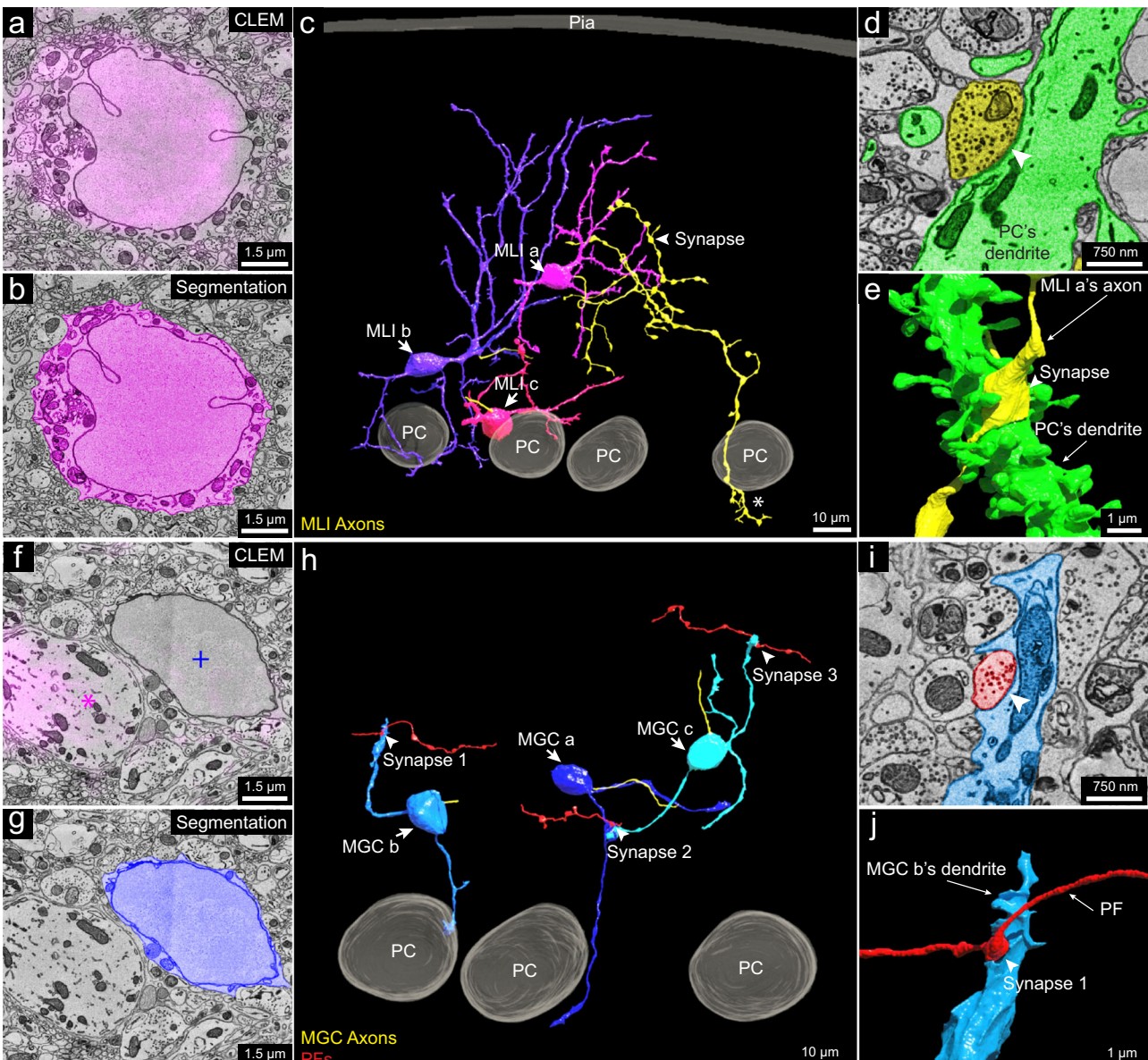

**Fig. 5 | 3D reconstruction of molecular layer interneurons and granule cells distinguished by PV-specific scFv probes. a** Representative 2D CLEM image (*n* = 22) showing fluorescence signals (magenta) of the parvalbumin-specific scFv probe overlapping a molecular layer interneuron (MLI) cell body. **b** EM image showing 2D segmentation (magenta) of the MLI in (**a**) (*n* = 1). **c** 3D reconstruction of MLIs (MLI a, MLI b, MLI c) (*n* = 3) relative to the pia and Purkinje cell layer, with axons in yellow. MLI a, farthest from the Purkinje cell layer, has an axon branching extensively and innervating a Purkinje cell's dendritic shaft at arrowhead (dendritic shaft not shown; the synapse is shown in **d** and **e**) and forming part of the pinceau structure around another Purkinje cell's axon initial segment at asterisk (axon initial segment not shown; the pinceau is shown in Supplementary Fig. 27e). This suggests MLI a is a deep axon stellate cell. **d** EM image showing 2D segmentation of the synapse (arrowhead) between MLI a's axon (yellow) and a Purkinje cell dendrite (green) (*n* = 1). **e** 3D reconstruction of the synapse in (**d**) (*n* = 1). **f** Representative 2D CLEM image showing fluorescence signal (magenta) of the PV-specific scFv probe overlapping Purkinje cell dendrites (magenta asterisk) but not labeling a molecular layer granule cell (MGC) (blue plus sign) (*n* = 7). **g** EM image showing 2D segmentation (dark blue) of the MGC cell body in (**f**) (*n* = 1). **h** 3D reconstructions of MGCs (MGC a, MGC b, MGC c) (*n* = 3) relative to three Purkinje cells. Unlike granule cells in the granule cell layer, these MGCs received synapses from parallel fibers. Three such synapses between parallel fibers (red) and the MGCs are labeled. **i** EM image showing 2D segmentation of synapse 1 in (**h**) (*n* = 1). **j** 3D reconstruction of synapse 1 (*n* = 1) labeled by the arrowhead in (**i**). *n* indicates an example.

terminal axonal arborizations in the granule cell layer innervating granule cells (Fig. 6k and Supplementary Fig. 28c), indicating that they are mossy fibers. The remainder of the axons had a similar appearance ultrastructurally but ran out of our EM volume, which prevented an unambiguous identification of their type, but they could also be mossy fibers. None of these axons were calbindin-positive. Axons that we suspect are primary Purkinje cell axons or their branches (because they were calbindin-positive; Supplementary Fig. 25a, inset 2) did not show $K_v1.2$ labeling at their juxtaparanodes. We thus conclude that at least a portion of the axons with $K_v1.2$-positive juxtaparanodes are mossy fibers, but not Purkinje cell axons.

VGluT1: VGluT1 is a member of the vesicular glutamate transporter family that is expressed by axonal terminals[53]. In the cerebellar cortex, VGluT1 is highly expressed in the boutons of parallel fibers and in a subgroup of mossy fiber arborizations[53,54]. In Crus 1, both in a previous study[54] and in our immunofluorescence images, there are at least three types of mossy fiber arborizations in the granule cell layer:

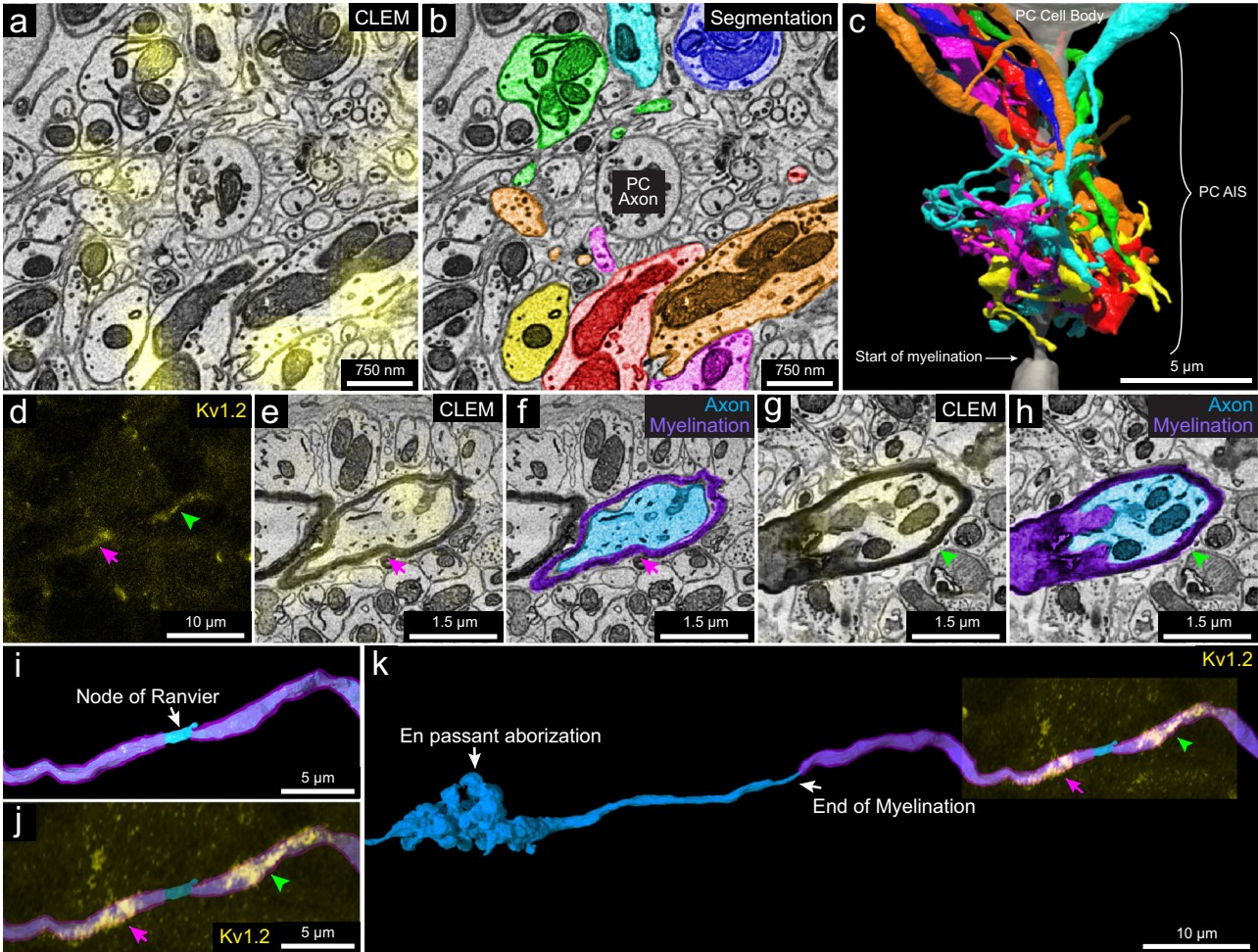

**Fig. 6 | 3D reconstruction of subcellular components labeled with the K$_v$1.2-specific scFv probe. a** 2D CLEM image showing the fluorescence signals (yellow) of the K$_v$1.2-specific scFv probe overlapping with axonal terminals of MLIs at a pinceau structure surrounding the axon initial segment of a Purkinje cell ($n = 1$). **b** EM image showing the 2D segmentations of some ($n = 7$) of the axon terminals labeled in (**a**). **c** 3D reconstruction of the seven axonal terminals in (**b**) (see Supplementary Fig. 28a for the morphology of these axonal terminals). **d** A representative pair of juxtaparanodal punctate labeling sites of the K$_v$1.2-specific scFv probe in the granule cell layer ($n = 15$). The magenta arrow and green arrowhead show each side of the juxtaparanodal labeling. **e** Representative 2D CLEM image ($n = 15$ examples) showing one side of the juxtaparanodal punctate labeling (yellow, magenta arrow) of the axon in (**d**). **f** Shows the 2D segmentation of the axon in (**e**) ($n = 1$). **g** Representative 2D CLEM image ($n = 15$) showing another side of the juxtaparanodal punctate labeling (yellow, green arrowhead) of the K$_v$1.2-specific scFv probe at the other side of the juxtaparanodal segment of the same node of Ranvier. **h** EM image showing the 2D segmentation of the axon in (**g**) ($n = 1$). **i** 3D reconstruction of the node of Ranvier of the axon labeled in (**d**–**h**) (myelination labeled in purple). **j** The juxtaparanodal labeling by the K$_v$1.2-specific probe in (**d**) overlaid onto the reconstructed node of Ranvier. **k** Extended 3D reconstruction of the axon in (**i**) shows that it had an *en passant* arborization and another terminal arborization (not shown) in the granule cell layer, that appeared to be mossy fiber synapses. *n* indicates an example.

VGluT1-positive, VGluT2 positive, and VGluT1/2 double positive terminals (Fig. 7a, c and Supplementary Fig. 29a, b-1–3). In our vCLEM dataset, the labeling of anti-VGluT1 scFv in the granule cell layer (cyan fluorescence signal in Fig. 7 a) does not label all mossy fibers (Fig. 7c). We classified the mossy fiber arborizations based on whether they were VGluT1-positive (Fig. 7a, b) or VGluT1-negative (Fig. 7c, d). While we can't further subdivide the VGluT1-positive from the VGluT1/2 double positive types, we surmise that the VGluT1-negative terminals are VGluT2-positive. To see if these two categories had ultrastructural differences, we randomly chose ten VGluT1-positive and ten VGluT1-negative mossy fiber terminals and reconstructed them. The 3D morphologies of both types were similar (Fig. 7e–g). However, the mean volumes of VGluT1-positive terminals were significantly larger than VGluT1-negative terminals ($P = 0.0191$; $P < 0.05$; $t = 2.575$, df = 18) (Fig. 7h) (see Supplementary Tables 5 and 6 for the volumes of each terminal). We also performed automatic detection of synaptic vesicles (Supplementary Fig. 29c, e) and

mitochondria in these terminals (Supplementary Fig. 29d, f) using machine learning algorithms (see Supplementary Tables 5 and 6 for the vesicle number, vesicle density, mitochondrial volume, and the mitochondria per terminal volume of each terminal). The mean value of the number of synaptic vesicles per terminal was significantly higher in VGluT1-positive terminals than in VGluT1-negative terminals ($P = 0.0050$; $P < 0.01$; $t = 3.200$, df = 18) (Fig. 7i). The mean value of the synaptic density per terminal trended higher in VGluT1-positive terminals than in VGluT1-negative terminals ($P = 0.0705$; $P < 0.1$; $t = 1.922$, df = 18) (Fig. 7j). The mean value of the mitochondria per terminal volume was not significantly different ($P = 0.1285$; $P > 0.1$; $t = 1.593$, df=18) (Fig. 7k). These results suggest that these two types of mossy fiber terminals have distinct EM ultrastructure. They may originate from different brain regions and have different electrophysiological properties. However, we found a number of granule cells that were innervated by both VGluT1-positive and VGluT1-negative mossy fibers.

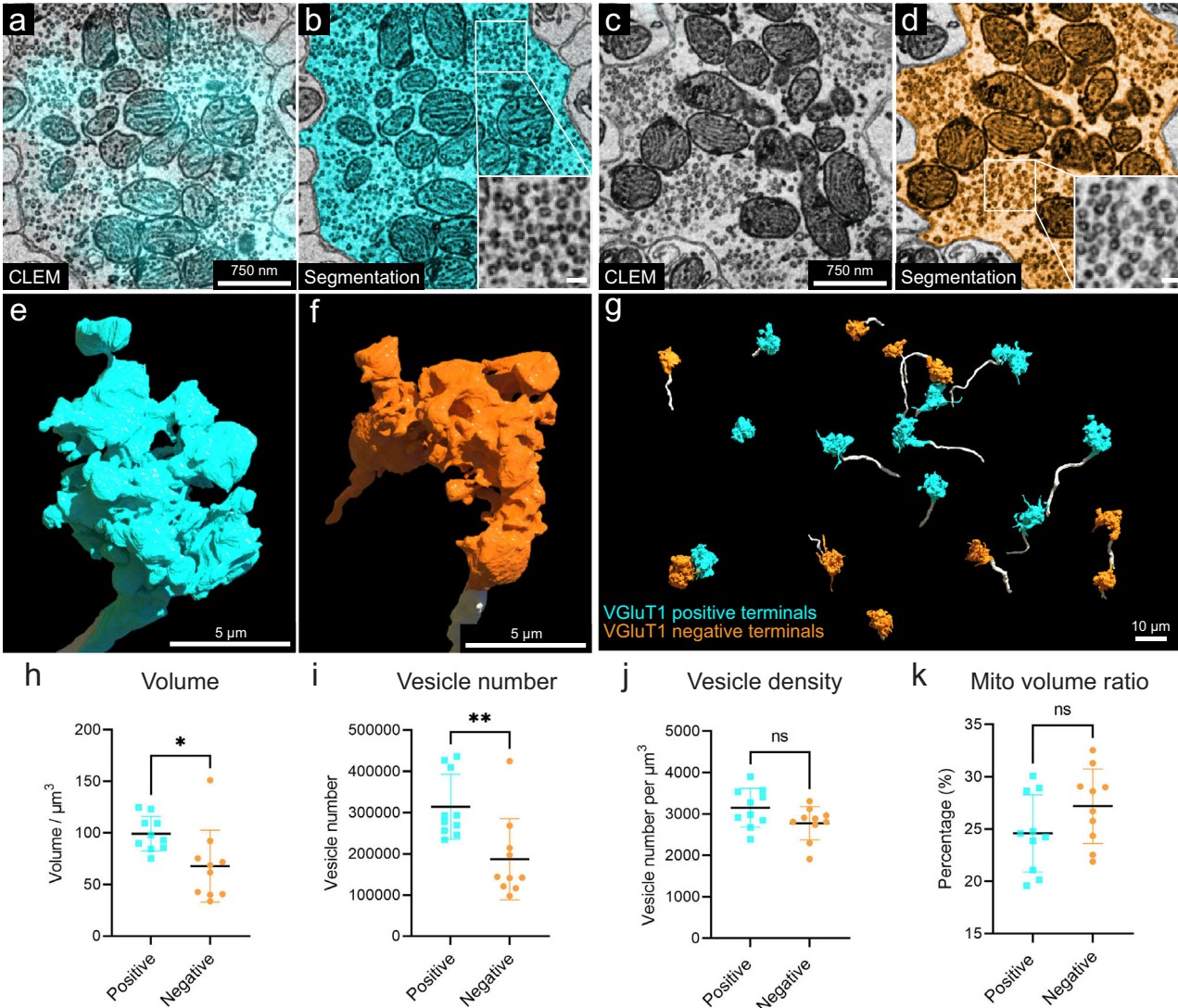

**Fig. 7 | 3D reconstruction and analysis of VGluT1-positive and -negative mossy fiber terminals based on labeling with the VGluT1-specific scFv probe.**
**a** Representative 2D CLEM image showing the fluorescence signals (cyan) of the VGluT1-specific scFv probe overlapping with a mossy fiber terminal ($n = 10$). **b** EM image showing the 2D segmentation (cyan) of the mossy fiber terminal labeled in (**a**) ($n = 1$). The enlarged inset shows the morphology of synaptic vesicles in this terminal. **c** Representative 2D CLEM image showing a mossy fiber terminal lacking VGluT1 fluorescence signal ($n = 10$). **d** EM image showing the 2D segmentation (orange) of the mossy fiber terminal labeled in (**c**) ($n = 1$). The enlarged inset shows the morphology of synaptic vesicles in this terminal. **e** 3D reconstruction of the VGLUT1-positive mossy fiber terminal (cyan) in (**b**). The axon leading to the terminal arises from the bottom of the panel. **f** 3D reconstruction of the VGluT1-negative mossy fiber terminal (orange) in (**c**). The axon leading to the terminal arises from the bottom of the panel. **g** 3D reconstruction of ten VGluT1-positive mossy fiber terminals (cyan) and ten VGluT1-negative mossy fiber terminals (orange). The axons are labeled in white. **h** Volume, **i** vesicle number, **j** vesicle density, **k** mitochondria volume ratio per terminal volume were measured for each of the 10 VGluT1-positive terminals ($n = 10$), cyan, and the 10 VGluT1-negative terminals ($n = 10$), orange. Mean values with SD are shown on each graph. Two-sided unpaired $t$ test was performed. For volume (**h**), $P = 0.0191$; $P < 0.05$; $t = 2.575$, df = 18; For vesicle number (**i**), $P = 0.0050$; $P < 0.01$; $t = 3.200$, df = 18; For vesicle density (**j**), $P = 0.0705$; $P < 0.1$; $t = 1.922$, df = 18; For mitochondria volume ratio (**k**), $P = 0.1285$; $P > 0.1$; $t = 1.593$, df = 18. Mito mitochondria. $n$ indicates example. Source data are provided as a Source Data file.

## Discussion

Here, we report a technique that visualizes multiple molecular labels superimposed on an EM volume, using detergent-free scFv-based immunofluorescence. We used this method to study a volume of cerebellum labeled with five scFv probes, acquired by vCLEM. This vCLEM dataset provided several insights relevant to the cytology of the cerebellar cortex. We deposited the expression plasmids of eight scFv probes in the open-access repository Addgene (see Table 1 for the links to the plasmid page at Addgene). Each of these probes possesses a sortase tag to allow straightforward attachment of fluorescent or other labels for use in vCLEM studies or other applications.

Expanding this technique to more probes will require acquiring a larger collection of functional scFvs to include additional commonly used molecular markers in bioscience. The scFv probes we generated were all based on mAbs from the UC Davis/NIH NeuroMab Facility[21] which recently generated more than 180 scFvs for common neuroscience-related targets[25], all of which are available at Addgene and many additional ones are in preparation. To generate more scFvs, several strategies present themselves. First, sequencing of mAb-encoding hybridoma cell cDNA[19] or de novo amino acid sequencing of a mAb[55] can establish the sequences of the $V_H$ and $V_L$ domains of other commonly used mAbs; the latter approach can be used for individual components of polyclonal Abs for subsequent conversion into scFvs[56]. Second, artificial protein evolution using phage-display methods has been used to modify the structure of failed scFvs to render them functional[57]. Third, rational protein design via point-

mutation or CDR-grafting has been used to improve the performance of failed scFvs[58,59]. Finally, purely in vitro approaches using phage display or other methods can be used to screen naive or immune scFv libraries to develop new scFvs[60]. We believe it is likely that there will be a much broader selection of scFvs available through open-access sources that can be effectively used as immuno-probes in the near future, and they will become more widely used for both conventional immunolabeling and volumetric CLEM.

Multiplexability, i.e., the ability to use multiple labels with different colors, can make vCLEM more cost-effective and more powerful. More resolvable color labels in a specimen means that more molecular information can be extracted without the need of multiple samples. Moreover, combinatorial labels may parse more cell types than possible with only a single label at a time. For example, in the cerebellum, cells that are CB- and PV-positive are Purkinje cells, whereas PV-positive, but CB-negative cells, are molecular layer interneurons. This same combinatorial approach will likely be useful in parsing various sub-types of interneurons in the cerebral cortex.

With spectral unmixing, we were able to acquire six-color fluorescent image volumes from samples immunolabeled with scFv probes. However, six-color images are not the upper limit of how many colors can be disambiguated. With the growing list of functional scFvs, optimized algorithms of linear unmixing can achieve super-multiplexed fluorescence imaging with up to 45 colors[13]. It is possible that scFvs could also be conjugated with narrow-peaked Raman imaging dyes to achieve multiplexed Raman vibrational imaging with 24 or more colors[61]. In addition, as our technique is based on immunolabeling, it can be easily combined with other types of functional vCLEM. For example, scFv-enabled immunofluorescence can be applied to samples that have been functionally imaged with calcium or voltage indicators, so that the molecular identities of the neurons whose functional properties have been analyzed can also be determined. This advantage can greatly expand the knowledge of neural networks examined by functional live imaging.

We used confocal microscopy to acquire fluorescence volumes of scFv-labeled samples because of the power of spectral unmixing to separate multicolor fluorescence. The superior ability of scFvs to penetrate 1-mm thick brain tissue sections as demonstrated in this study can allow large-scale vCLEM with techniques like multicolor 2-photon microscopy[62–65], lightsheet microscopy in uncleared tissue[66,67], and confocal microscopy done with clearing approaches compatible with electron microscopy[68].

One of the fundamental questions in neuroscience is the relationship between transcriptomically derived cell types and histologically derived cell classifications. The use of CLEM with many molecular probes is a path forward. One area is the use of machine learning to differentiate two different molecularly defined cell types. We showed this when distinguishing between two kinds of mossy terminals, despite having the same neurotransmitter and being functionally and gross-anatomically similar (Fig. 7). More recent machine learning algorithms like "SegCLR"[69] may also be a promising approach for cell typing in vCLEM datasets.

## Methods

### Animals
Animals used in the study were adult (8–12 weeks) female C57BL/6J and YFP-H mice (Jackson Laboratory #003782) or adult (8–12 weeks) female Sprague Dawley rats. The vCLEM dataset was collected from a female adult C57BL/6J mouse. All experiments using mice were conducted according to US National Institutes of Health guidelines and approved by the Institutional Animal Care and Use Committee at Harvard University (protocol number 24-08-4). The housing conditions are: Dark/light cycle: Lights on at 6 am and off at 8 pm (14 L:10D); Ambient temperature: 22 °C ± 1; Humidity: 30–70%. All experiments for rat immunohistochemistry were performed in strict compliance with the University of California Davis ethical regulations for studies involving animals as approved by the University of California Davis Animal Care and Use Committee (protocol #: 23734). UC Davis Animal Welfare Assurance Number (Vertebrate Animals) A-3433-01. The housing conditions are: Dark/light cycle: Lights on at 7 am and off at 7 pm (12 L:12D); Ambient temperature: 20–26 °C; Humidity: 30–70%.

### ScFv production
The amino acid sequence of scFv was designed by linking the variable domains of the heavy chain and the light chain of a mAb with a (GGGGS GGGGS GGGGS) linker (3× linker), a (GGGGS GGGGS GGGGS GGGS) linker (4× linker 1), or a (GGGGS GGGGS GGGGS GGGGS) linker (4× linker 2). A signal peptide (MDWTWRILFLVAAATGAHS or MGWSCIILFLVATATGVHS) was added to the N'-terminus. A sortase tag (SLPETGG) and a 6× His tag was added to the C'-terminus. The amino acid sequence was reverse-translated into a DNA sequence with codon optimization for human cells. The DNA sequence was synthesized and cloned into pcDNA 3.1 or pcDNA 3.4 vectors.

Expi 293 cells (Thermo Fisher #A14527) were used for scFv expression. The cells were maintained in Expi 293 Expression Medium (Thermo Fisher # A1435101) and passaged every other day. After two rounds of passage, transfection of the scFv expression plasmids was carried out using 293fectin (Thermo Fisher #12347019) following the vendor's protocol. For 100 ml culture with $1 \times 10^8$ cells, we used 100 μg plasmid DNA and 200 μl 293fectin. Cells were incubated for another 7 days, after which the culture supernatant was collected. The supernatant was incubated with Ni-NTA agarose (Thermo Fisher # R90110). After washing with Tris buffer (50 mM Tris, 150 mM NaCl, pH = 7.5–8.0), the bound proteins were eluted with the elution buffer (250 mM imidazole, 50 mM Tris, 150 mM NaCl, pH = 7.5–8.0). The eluted proteins were loaded into Millipore Sigma Amicon Ultra-15 centrifugal filter unit (Fisher Scientific #UFC903024), washed with Tris buffer, and then buffer exchanged into Tris buffer with 15% glycerol (Millipore Sigma #G5516) by centrifugation at 3257×g at 4 °C. The concentration of the purified protein was determined by A280. The purified protein was stored at −20 °C.

### Sortase reaction
The GGGC peptide was synthesized at BIOMATIK. The GGGC peptides were dissolved in 1× PBS (Millipore Sigma #P3813) at a concentration of 20 mg/ml. The fluorescent dyes in maleimide form (Thermo Fisher # A10254, # A20347, # A10256, # A10255; Lumiprobe # 17180) were dissolved in DMSO (Millipore Sigma # D8418) at the concentration of 20 mM. The dissolved peptide and fluorescent dyes were mixed at the molar ratio of 12:1 and then incubated with shaking at 800 rpm at 4 °C overnight on an Eppendorf ThermoMixer (same below). The GGGC-dye conjugates were purified by HPLC. The identity of the purified conjugates were confirmed by mass spectroscopy, lyophilized, and then re-dissolved in ddH2O at the concentration of 4 mM.

The sortase reaction was performed as described in ref. 70. In brief, GGGC-dye conjugates (500 μM) and sortase tag-containing scFvs (100 μM) were mixed in the sortase reaction buffer (50 mM Tris-HCl, 150 mM NaCl, 10 mM CaCl2, pH = 7.5–8.0), followed by adding sortase A pentamutant that has a 6× His tag (2.5 μM) (a gift by Dr. Tao Fang). The mixture was incubated with shaking at 500 rpm at 12 °C for 3 h. Then Ni-NTA agarose was added into the mixture, followed by an incubation shaking at 500 rpm at RT for 20 min to remove sortase and any leftover GGGC-dye conjugates. After the incubation, the mixture was loaded into a mini-column (USA Scientific #1415-0600) and centrifuged briefly to remove the Ni-NTA agarose. The filtered solution that contains the scFv-dye conjugates was loaded into Millipore Sigma Amicon Ultra-4 centrifugal filter unit (Fisher Scientific #UFC801024), washed with Tris buffer, and then buffer exchanged into Tris buffer with 15% glycerol by centrifugation at 3257 × g at 4 °C.

The concentration of the purified scFv-dye conjugates was determined by A280. The purified scFv-dye conjugates were stored at −20 °C.

**Perfusion and fixation**
The mouse was anesthetized by isoflurane until there was no toe-pinch reflex. Mice were then transcardially perfused with aCSF (125 mM NaCl, 26 mM NaHCO₃, 1.25 mM NaH₂PO₄, 2.5 mM KCl, 20 mM glucose, 1 mM MgCl₂ and 2 mM CaCl₂ (Millipore Sigma #S7653, S6297, S8282, P9333, G7528, M1028, 21115) at the flow rate of 10 ml/min for 2 min to remove blood, followed with 4% formaldehyde fresh prepared from paraformaldehyde (Electron Microscopy Sciences 15714), 0.1% glutaraldehyde (Electron Microscopy Sciences 16220) in 1× PBS (Millipore Sigma P3813) for 3 min for fixation. Brains were dissected and then post-fixed in the same fixative on a rotator overnight at 4 °C. Brains were sectioned into 50-μm or 120-μm coronal sections using a Leica VT1000 S vibratome and stored in the same fixative at 4 °C.

For ECS-preserving perfusion, the detailed protocol was described in ref. [17]. In brief, mice were anesthetized by isoflurane, transcardially perfused with aCSF at the flow rate of 10 ml/min for 2 min to remove blood, followed with 15 w/v% mannitol (Millipore Sigma M9546) aCSF solution for 1 min, 4 w/v% mannitol aCSF solution for 5 min, and 4 w/v% mannitol, 4% formaldehyde in 1× PBS for 5 min. Brains were dissected out and then post-fixed in the same fixative for 3 h on a rotator at 4 °C. Brains were sectioned into 50-μm coronal sections using a Leica VT1000 S vibratome, and then store in 1× PBS at 4 °C.

For Sprague Dawley rats, following deep anesthesia induced by administration of pentobarbital (Millipore Sigma # P3761), animals were transcardially perfused with 4% formaldehyde (freshly prepared from paraformaldehyde, Millipore Sigma #158127) in 0.1 M sodium phosphate buffer pH 7.4 (0.1 M PB). Sagittal brain sections, 30 μm-thick, were prepared and immunolabeled using free-floating methods as described below and detailed previously[71].

**Brain section fluorescence immunohistochemistry**
Detergent-free immunofluorescence labeling was performed with scFv probes or nanobody probes (see Supplementary Table 1 for the scFv and nanobody probes used in this study). In brief, 50-μm or 120-μm coronal sections were transferred into 3.7-ml shell vials (Electron Microscopy Sciences 72631-10) with 1 ml 1× PBS. The sections were first washed with 1× PBS for 3 × 10 min, and then blocked in glycine blocking buffer (0.1 M glycine (Millipore Sigma #50046), 0.05% NaN₃ (Millipore Sigma #S8032) in 1× PBS) for 1 h on a rotator at 4 °C. The labeling solution was prepared by diluting scFv-dye conjugates or nanobody-dye conjugates in glycine blocking buffer (see Supplementary Table 1 for the final concentration of each scFv and nanobody probe). The labeling solution and any remaining steps were protected from light. The sections were then incubated with the labeling solution on a rotator at 4 °C. In all, 50-μm sections were incubated for 3 days. In all, 120-μm sections were incubated for 7 days. After the incubation, sections were washed with 1× PBS for 3 × 10 min, and then stained with Hoechst 33342 (Thermo Fisher #H3570, diluted 1:5000 in 1× PBS) for 1 h on a rotator at 4 °C. Sections were washed with 1× PBS for 3 × 10 min, and then mounted onto glass slides (VWR # 48311-703) with Vectashield H-1000 (Vector Laboratories) or with 1× PBS (for vCLEM).

Immunofluorescence labeling with detergent was performed with primary antibodies plus secondary antibodies for single labeling (for the validation of scFv probes versus mAbs or pAbs; see Supplementary Figs. 12, 13, and 15), or with primary antibodies plus secondary antibodies and scFv probes for double labeling (for the double labeling experiment of VGluT1 and VGuT2; see Supplementary Fig. 29) (see Supplementary Table 2 for the primary antibodies and Supplementary Table 3 for the secondary antibodies used in this study). The detailed protocol is available on the NeuroMab website. In brief, 50-μm coronal sections were transferred into 3.7-ml shell vials with 1 ml 1× PBS. The

sections were first washed with 1× PBS for 3 × 10 min, and then blocked in the vehicle (10% normal goat serum (Abcam #ab7481), 0.3% Triton X-100 (Millipore Sigma #T9284) in 1× PBS) overnight on a rotator at 4 °C. Subsequent steps were protected from light. For unconjugated mAbs or pAbs, the sections were then incubated with the primary antibody (mAbs or pAbs) solution or primary antibody solution plus scFv probes (for the experiment of the double labeling of VGluT1 and VGuT2; see Supplementary Fig. 29) (see Supplementary Tables 1 and 2 for the dilution ratios of primary antibodies and the anti-VGluT1 scFv probe used in this study) for 2 days on a rotator at 4 °C. After the incubation, sections were washed with vehicle for 3 × 10 min, and then incubated with the secondary antibody solution (see Supplementary Table 3 for the dilution ratios of secondary antibodies used in this study) overnight on a rotator at 4 °C. For dye-conjugated mAbs (for experiments in Supplementary Fig. 15), sections were incubated with dye-conjugated mAbs diluted in the vehicle on a rotator at 4 °C for 7 days. After the incubation, sections were washed with 1× PBS for 3 × 10 min, and then stained with Hoechst (diluted 1:5000 in 1× PBS) for 1 h on a rotator at 4 °C. Sections were washed with 1× PBS for 3 × 10 min, and then mounted with Vectashield H-1000 onto glass slides.

For experiments with rat sections, sections were permeabilized and blocked in 0.1 M PB containing 10% goat serum and 0.3% Triton X-100 (vehicle) for 1 h at RT, then incubated overnight at 4 °C in primary antibodies diluted in the vehicle. After four 5-min washes in 0.1 M PB, sections were incubated with mouse IgG subclass- and/or species-specific Alexa-conjugated fluorescent secondary antibodies and 10 μM Hoechst 33258 DNA stain (Thermo Fisher# H2149) diluted in the vehicle at room temperature (RT) for 1 h. After two 5-min washes in 0.1 M PB followed by one 5-min wash in 0.05 M PB, sections were mounted and air dried onto Superfrost plus microscope slides (Fisher Scientific #12-550-15) and mounted with Prolong Gold (Thermo Fisher # P36930).

**Cell culture immunofluorescence immunocytochemistry**
COS-1 cells (ATCC cat# CRL-1650) were maintained in Dulbecco's modified Eagle's medium (Gibco #11965118) supplemented with 10% bovine calf serum (HyClone # SH30073.03), 1% penicillin/streptomycin (Gibco # 15140-122, and 1× GlutaMAX (Gibco # 35050061). HEK293T cells (ATCC cat# CRL-3216) were maintained as COS-1 cells with the exception that 10% or 5% Fetal Clone III fetal calf serum (HyClone # SH30080.03) was used in place of bovine calf serum. COS-1 and HEK293T cells were seeded onto 22-mm² poly-D-lysine (Thermo Fisher # A3890401) coated coverslips (10⁵ cells/coverslip) in six-well cultures plates or into glass-coverslip bottom 3.5-mm dishes (ibidi #81218-200). After adherence, cells were transfected with mammalian expression plasmids encoding mEmerald-tagged GFAP (Addgene #54107 or Flag-tagged human calbindin (Origene # RC201358) using Lipofectamine 2000 (Thermo Fisher # 11668500) or Lipofectamine 3000 (Thermo Fisher # L3000001) transfection reagent following the manufacturer's protocol. Cells were used 40–48 h post-transfection.

For experiments in Supplementary Fig. 7, calbindin-transfected HEK293T cells cultured on glass-coverslip bottom 3.5-mm dishes, were washed with 1× PBS and then fixed in the same buffer containing 4% formaldehyde prepared fresh from paraformaldehyde + 0.1% glutaraldehyde for 15 min at RT. After washing with 1× PBS for 3 × 10 min, cells were blocked in glycine blocking buffer (0.1 M glycine, 0.05% NaN₃ in 1× PBS) for 1 h on a rotator at RT. The labeling solution was prepared by diluting scFv-dye conjugates in glycine blocking buffer or vehicle (10% normal goat serum, 0.3% Triton X-100 in 1× PBS) (see Supplementary Table 1 for the final concentration). The labeling solution and any remaining steps were protected from light. The cells were then incubated with the labeling solution on a rotator at RT overnight. After the incubation, cells were washed with 1× PBS for 3 × 10 min, and then stained with Hoechst 33342 (Invitrogen, diluted

1:5000 in 1× PBS) for 1 h on a rotator at 4 °C. For the live cell labeling, cells were washed with 1× PBS and then incubated in culture media with diluted scFv-dye conjugates (same concentration as experiment with fixation) for 1 h at 37 °C. After the incubation, cells were washed with 1× PBS for 10 min, and then fixed in 1× PBS containing 4% formaldehyde prepared fresh from paraformaldehyde + 0.1% glutaraldehyde for 15 min at RT, washed with 1× PBS for 3 × 10 min, and then stained with Hoechst 33342 (Invitrogen, diluted 1:5000 in 1× PBS) for 1 h on a rotator at 4 °C. After a final 3 × 10 min wash in 1× PBS, the dishes were placed onto a Zeiss LSM 880 confocal microscope for imaging.

For experiments in Supplementary Figs. 8 and 9, immunofluorescence labeling of transfected COS-1 and HEK293T cells was performed using two different protocols. For immunolabeling GFAP-transfected cells, were washed three times in ice-cold PBS (10 mM phosphate buffer, pH 7.4, 0.15 M NaCl) containing 1 mM $MgCl_2$ and 1 mM $CaCl_2$ and then fixed in the same buffer containing 3% formaldehyde prepared fresh from paraformaldehyde plus 0.1% Triton X-100 for 30 min at RT. After three washes with PBS, nonspecific protein binding sites were blocked with Blotto (4% nonfat dry milk powder in Tris-buffered saline (10 mM Tris-HCl, pH 7.5, 0.15 M NaCl) plus 0.1% Triton X-100 (Blotto-T) for 1 h at RT and then incubated with either 5-TAMRA-labeled anti-GFAP N206B/9 scFv or anti-GFAP N206A/8 mouse IgG1 mAb for 1 h at RT. After washing three times with Blotto-T, mAb labeled cells were incubated with Biotium 770 nm goat anti-mouse IgG1 secondary antibody diluted in Blotto-T containing 0.8 μg/ml Hoechst 33258 DNA stain (Thermo Fisher # H1398) for 1 h, and washed three times with PBS containing 0.1% Triton X-100.

A similar protocol was used to immunolabel permeabilized calbindin-transfected cells, except that primary antibodies were Alexa 594 anti-calbindin scFv or anti-calbindin L109/39 mouse IgG2b mAb, and Alexa 594 goat anti-mouse IgG2b secondary antibody were used. For labeling non-permeabilized calbindin-transfected cells, the protocol differed in that cells were fixed in PBS containing 3% formaldehyde prepared fresh from paraformaldehyde for 30 min at RT in the absence of Triton X-100. After three washes with PBS, nonspecific protein binding sites were blocked with Blotto (no Triton X-100) for 1 h at RT and then incubated with either the anti-calbindin scFv or mAb as above for 1 h at RT. After washing three times with Blotto (no Triton X-100), cells were permeabilized with Blotto containing 0.1% Triton X-100 (Blotto-T) for 1 h at RT. Cells were then incubated with anti-Flag rabbit polyclonal antibody for 1 h at RT. Cells were washed three times in Blotto-T, incubated with Alexa 488 goat anti-rabbit IgG (H + L) and for mAb labeled cells also with Alexa 594 goat anti-mouse IgG2b secondary antibody diluted in Blotto-T containing 0.8 μg/ml Hoechst 33258 DNA stain for 1 h, and washed three times with PBS containing 0.1% Triton X-100. Coverslips were mounted on microscope slides using Prolong Diamond.

### Fluorescence confocal microscopy

In all, 50-μm sections were mounted in Vectashield H-1000 with a #1.5 coverslip (Electron Microscopy Sciences #72204-01) on top sealed with clear nail polish. 120-μm sections (for vCLEM) were mounted in 1× PBS inside 120-μm spacer (Thermo Fisher #S24737) with a #1.5 coverslip on top sealed with the spacer. In all, 50-μm sections and HEK293T cell in glass-coverslip-bottom dishes (in Supplementary Fig. 7) were imaged with a Zeiss LSM 880 confocal laser scanning microscope equipped with either a 20×/0.8 NA air-objective, or a 40x/1.1 NA water-immersion objective, or a 63×/1.4 NA oil-immersion objective depending on the required optical resolution. Acquisition of double or triple-color fluorescent images was done with appropriate band-pass filters for the specific fluorescent dyes to avoid crosstalk.

Images of rat brain sections were taken using a Hamamatsu C11440 sCMOS camera installed on an AxioObserver Z1 microscope with a 10×/0.5 NA lens, Zeiss Collibri 7 LED light source, and an ApoTome2 coupled to Zen software (Zeiss, Oberkochen, Germany). Where

appropriate (e.g., when comparing scFv and the mAb from which it was derived), the same exposure time was used to compare images directly.

Labeled COS-1 or HEK293t cells (in Supplementary Figs. 8 and 9) were imaged under indirect immunofluorescence on a Zeiss Axioskop2 microscope using optical sectioning with Apotome2 structured illumination and a 63x/1.4 NA objective. All images of calbindin labeling (scFv, mAb, and anti-Flag) were captured using Zen Pro using the same exposure times for permeabilized and non-permeabilized cells. For presentation, images were exported as TIFFs, linearly scaled for min/max intensity, and flattened as RGB TIFFs in Photoshop (Adobe).

For spectral imaging, the detailed protocol was described in ref. 30. In brief, 120-μm sections immunolabeled by five scFv with different fluorescent dyes (Alexa fluor 488, 532, 594, 647, and 5-TAMRA) were imaged with a Zeiss LSM 880 confocal laser scanning microscope equipped with a 40×/1.1 NA water-immersion objective. All fluorophores except Hoechst were excited simultaneously with 488, 560, 633 nm lasers and a 488/561/633 nm beam splitter. A z-stack of lambda stack images with the interval of 1 μm in z axis was acquired by an array of 32 Gallium Arsenide Phosphide (GaAsP) detectors in lambda mode at 9 nm bins from 414 nm to 691 nm. Lambda stack images were acquired from each of the sections labeled by only one of the five scFvs with the same settings in lambda mode. Reference spectra for each fluorescent dye were extracted from each of these lambda stacks. Linear unmixing was performed on the z-stack of lambda stack images acquired from the multi-labeled section using the reference spectra in Zen Blue software (Zeiss). The Hoechst channel (labeling cell nuclei) of the multi-labeled section was acquired separately with a single 405 nm laser and appropriate band-pass filters.

To test if the Hoechst channel could be combined into linear unmixing, we tried acquiring another z-stack of lambda stack images of another multi-labeled 120-μm section with simultaneous excitation with 405, 488, 560, 633 nm lasers, a 488/561/633 nm beam splitter and a 405 nm beam splitter. A lambda stack image was acquired from a section labeled by only Hoechst with the same settings in lambda mode. Linear unmixing was performed on the z-stack of lambda stack images of the multi-labeled section using the reference spectra (five fluorescent dyes plus Hoechst) in Zen Blue software (Zeiss). Several reference spectra for Hoechst extracted from different locations of pixels in Hoechst's lambda stack image caused the Hoechst signals to be unmixed into the channel of Alexa 488 in error, but we did find one spectrum that generated good unmixing results (Supplementary Fig. 19).

The brightness, contrast, and gamma of all fluorescent images were adjusted. Fluorescence image volumes were projected to a single plane by maximum intensity for visualization in 2D.

### ScFv, nanobody, and antibody penetration experiment

Immunofluorescence with or without detergent by scFv probes, nanobody probes, primary antibodies with dyes directly conjugated, or with primary antibodies plus secondary antibodies was performed on 300-μm or 1-mm coronal sections from animals perfused with normal or ECS-preserving protocol. See above for a detailed protocol of immunofluorescence. For the sections labeled with scFv probes, nanobody probes, primary antibodies with dyes directly conjugated, the incubation lasted for 1 day, or 7 days. For saponin-treated samples, 10% saponin (Thermo Fisher # A18820.22) was diluted in vehicle (no Triton-X) to the final concentrations of 0.05%, 0.1% or 0.2%. For Triton-X treated sample, Triton-X was diluted in vehicle (no Triton-X) to the final concentrations of 0.1% or 0.3%. For the sections labeled with the primary mAb (no dye-conjugated) for calbindin (Supplementary Fig. 2b), the sections were incubated with the primary antibody solution (see Supplementary Table 2 for the dilution ratio of this mAb) for 4 days on a rotator at 4 °C. After the incubation, sections were washed with vehicle for 3 × 10 min, and then incubated with the secondary

antibody solution (see Supplementary Table 3 for the dilution ratio of the secondary antibody) for 3 days on a rotator at 4 °C. After the incubation, sections were washed with 1× PBS for 3 × 10 min. Then the sections were sliced into 50-μm or 120-μm sections lengthwise (see Supplementary Fig. 1b for detail) using a VT1000 S vibratome. The section from the middle was mounted onto glass slides, and then imaged by confocal microscopy to evaluate the penetration depth.

## EM preparation

After 120-μm sections were imaged, a small amount of 1× PBS was added between the glass and coverslip to unseal the coverslip. Sections were transferred into 3.7-ml shell vials with 1 ml secondary fixative (2% formaldehyde, 2,5% glutaraldehyde in 0.15 M sodium cacodylate (Electron Microscopy Sciences #12300) buffer with 4 mM $Ca^{2+}$ and 0.4 mM $Mg^{2+}$) and incubated for at least one week on a rotator at 4 °C. A modified ROTO (Reduced Osmium-Thiocarbohydrazide-Osmium) protocol was used to stain the sections. All steps were carried out at room temperature (RT) except when specified. Sections were washed 3 × 10 min in 0.15 M sodium cacodylate buffer with 4 mM $Ca^{2+}$ and 0.4 mM $Mg^{2+}$, and then incubated in a solution containing 2 w/v% $OsO_4$ (Electron Microscopy Sciences #19190), 1.5% w/v potassium ferrocyanide (Electron Microscopy Sciences #20150) and 0.15 M cacodylate buffer inside 120-μm spacers (the top surfaces were not peeled) between a glass slide and a coverslip for 2 h with a change of the same solution after the first hour to prevents tissue deformation. Sections were incubated in the same solution in shell vials for another 1.5 h on a rotator. After washing 3 × 10 min in $ddH_2O$, sections were incubated in a filtered aqueous solution containing 1 w/v% thiocarbohydrazide (Electron Microscopy Sciences #21900) for 45 min. After washing 3 × 10 min in $ddH_2O$, sections were incubated in an aqueous solution containing 2 w/v% $OsO_4$ for 1.5 h. After washing 3 ×10 min in $ddH_2O$, sections were incubated in a filtered aqueous solution containing 1 w/v% uranyl acetate (Electron Microscopy Sciences #22400) overnight on a rotator protected from light. On the next day, after washing 3 ×10 min in $ddH_2O$, sections were dehydrated in a series of graded (25, 50, 75, 100, 100, 100 v/v%, 10 min each) acetonitrile (Electron Microscopy Sciences #10021) aqueous solutions. Afterward, sections were infiltrated with 25% LX-112 (Ladd Research #21212): acetonitrile for 1.5 h, 50% LX-112: acetonitrile for 3 h, 75% LX-112: acetonitrile overnight, 100% LX-112 for 12 h on a rotator. Then the sections were embedded in 100% LX-112 at the bottom of a flat bottom embedding capsule (Ted Pella # 132) and cured in a 60 °C oven for 2 days.

## X-ray MicroCT scanning

X-ray micro-computed tomography (μCT) images of the resin-embedded sections were acquired using a Zeiss Xradia 510 Versa system and Zeiss' Scout and Scan software. The imaging conditions include a projection pixel size of around 1.15 μm (pixel bin size of 1), 60 kV X-ray source, 30 s exposure per projection, and 3501 projections for 360 degrees of rotation. Each scan took ~33 h. The Feldkamp, Davis, and Kress (FDK) algorithm reconstructed 3-dimensional volume maintains the projection pixel size of 1.15 μm resulting in a voxel size of 1.15 μm³. The Scout and Scan Control System Reconstructor software was used to convert the reconstructed files to.tiff files.

## EM imaging

The resin-embedded sections were cut into 30 nm serial ultrathin sections using automated tape-collecting ultramicrotome (ATUM)[31,72]. Serial sections were collected onto carbon-coated and plasma-treated Kapton tape. The tape was cut into strips and affixed onto 150 mm silicon wafers (University Wafer ID857).

To evaluate the ultrastructure of samples untreated with detergents or treated with detergents (Triton X-100 or saponin), ultrathin sections at 10 μm from the sample surface were images for each sample to ensure consistent comparison.

A Zeiss Sigma scanning electron microscope was used to acquire images for ultrastructure evaluation or overview images from the serial sections. Two overview images were taken per wafer, which were around 1.35 μm apart in z axis. Typical imaging conditions are 8-kV landing energy, 1.2-nA beam current, 3-μs dwell time, 150-nm pixel size, and 4k × 4k images. The images were captured using a below-the-lens backscatter detector and Zeiss' Atlas 5 software. The overview images were aligned using the "Linear stack alignment with SIFT" plugin in FIJI.

Prior to acquiring high-resolution images, the serial section sections on wafers were post-stained for 4 min with a 3% lead citrate solution (Electron Microscopy Sciences #22410). After staining, the sections were degassed for a minimum of 24 h at 1×10-6 Torr. A Zeiss MultiSEM 505 scanning electron microscope equipped with 61 electron beams was used to acquire high-resolution images from the serial sections. Images were collected using a 1.5-kV landing energy, 4-nm image pixel, and a 400-ns dwell time.

## High-resolution EM image processing

The preparation of the ssEM data before it could be segmented/analyzed includes two steps: affine stitching and elastic alignment. Affine stitching was performed on the raw data coming from the microscope. First, SIFT features were extracted from the boundary area of each raw tile. Then all the neighborhood tiles in each section were matched together based on SIFT feature matching. Finally, a global optimization step made the matching results smooth. After each section was stitched well, an elastic alignment of all the sections was performed. A rough affine transformation between each mFov in section i to section i + 1 and i + 2 was estimated based on the SIFT features of blob-like objects detected in the sections. Then template matching, performed as a fine-grained matching for section i to section i + 1 and i + 2, was done on grid-distributed image blocks in section i. At last, a global optimization on all the sections in the whole stack made the elastic transformation of mesh grid points on all the sections smooth.

The aligned stack was rendered at full resolution (4 × 4 × 30 nm) and each section was cut into 4k × 4k.png tiles, imported into VAST as a.vsvi file, and ingested into Neuroglancer for further analysis.

## Co-registration of fluorescence and EM volumes

The EM volume at a downsampled pixel size of 128 nm in the x and y plane and 480 nm in the z axis was exported from the high-resolution.vsv file in VAST. The multicolor fluorescent image volume's brightness and contrast were adjusted with the "Stack contrast adjustment" plugin in FIJI to compensate for signal decay in the z axis. The pixel sizes in the x and y plane and in the z axis were resampled to match those of the confocal fluorescence image volume. Both volumes were loaded into FIJI. With the BigWarp plugin, 188 landmark points were manually placed on corresponding sites of blood vessels, cell nuclei, cell bodies, and axons in the two volumes. The fluorescent image volume was 3D transformed by a thin plate spline interpolation based on the point correspondences. The transformed fluorescence volume was resampled in the z axis and then imported into VAST as.vsv files to be overlaid with the EM volume at a downsampled pixel size of 128 nm in the x and y plane and 30 nm in the z axis. The.vsv files were also ingested into Neuroglancer and resampled in the z axis to be overlaid with the EM volume at a pixel size of 8 nm in the x and y plane and 30 nm in the z axis.

## Automatic segmentation

The vEM dataset was segmented in 3D using Flood-Filling Networks (FFNs)[35]. The FFN segmentation model was trained at 32 × 32 × 30 or 16 × 16 × 30 nm resolution on the H01 dataset[36], and run here on CLAHE intensity normalized data[73] downsampled to match the trained model

resolution. The resulting base supervoxels were assembled into larger per-cell segments via manual proofreading.

The vEM dataset was also segmented in 2D at $8 \times 8 \times 30$ nm resolution by a method developed in our lab. A description of this approach is given in refs. 37,74. In brief, a pretrained algorithm was used to generate ground truth tiles of membrane predictions, which was corrected by a human annotator. Three rounds of ground truth correction were used to iteratively train deep neural network[38] with a UNET architecture[75]. Neural network predictions were done on a commodity GPU using MATLAB connected to a VAST[76] instance, serving the EM images. In each round the model was applied on tiles randomly selected from the entire EM space and corrections were made to tiles that contained errors. Producing further training rounds halted when a sufficient accuracy was met. Finally, 2D segmentation using a region-growing algorithm was applied on the entire space according to the local minima of the membrane predictions.

Human annotators manually assembled and proofread the segmentations by FFN to 3D reconstruct cells and cellular structures for analysis.

### Mitochondria detection
We implemented an anisotropic U-Net architecture, incorporating a combination of 2D and 3D convolutions, to predict binary mitochondrial masks and instance maps similar to the U3D-BC approach in MitoEM[77]. Given that the annotations for the new training dataset are confined to a select number of mossy terminals rather than densely annotated, our model exclusively considers regions containing labeled terminals for loss computation. This strategy prevents the generation of empty predictions for valid mitochondria solely due to the absence of annotations within labeled terminals. To separate individual mitochondria, including those in close proximity, we employ marker-controlled watershed segmentation based on the network's predictions. The segmentation system is implemented with the open-source PyTorch Connectomics deep learning toolbox[78].

### Vesicle segmentation and visualization
We first saturated and segmented all vesicles on six images from a terminal, totaling around 8,400 instances. Then, we fine-tuned the Cytoplasm 2.0 model from Cellpose[79,80] on these annotations. Specifically, as the model was pretrained on images with cells of a bigger size, we scaled up our image and segmentation maps of vesicles by $2\times$ in both XY dimensions. For the 20 volumes of terminals, we obtained roughly 5 million vesicle instances.

### Statistics and reproducibility
For Figs. 3–6 and Supplementary Figs. 25–29, all the fluorescence overlays and reconstructions were performed on 848 slices of EM images in the vCLEM dataset, and $n$ indicates an example.

After the brightness of contrast of the layer of the fluorescent signals of VGluT1 were fixed at a value in Neuroglancer, a human annotator randomly picked 10 locations of VGluT1-positive mossy fiber terminals and 10 locations of VGluT1-negative mossy fiber terminals on slice 512 of the vEM dataset. After 3D reconstruction of these terminals, the volume size of each terminal was generated using VAST. Mitochondria and synaptic detections (see above) were performed on segmentation of each terminal. None of these approaches were done blind to whether a terminal is VGluT1-positive or negative. Two-tailed, unpaired $t$ tests on volume, synaptic vesicle number, synaptic density, and mitochondria volume per terminal were performed in Prism-GraphPad.

### Reporting summary
Further information on research design is available in the Nature Portfolio Reporting Summary linked to this article.

## Data availability
The DNA sequences of the scFvs generated in this study have been deposited in GenBank under accession codes PP840912-PP840919. The expression plasmids of the scFvs generated in this study have been deposited at Addgene: https://www.addgene.org/browse/article/28238227/. The vCLEM dataset and its segmentation are publicly available through the Neuroglancer link in the source data file. The authors declare that all the representative data supporting the findings of this study are available within the paper and its supplementary information files. Other data that support the findings of this study are available from the corresponding authors upon request. Source data are provided with this paper.

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

## Acknowledgements

The authors thank the Harvard Center for Biological Imaging (RRID:SCR_018673) for infrastructure and support, D. Richardson at Harvard Center for Biological Imaging for advice on linear unmixing, the Biopolymers and Proteomics Core Facility at the Koch Institute at MIT for processing the peptide-fluorescent dye conjugates, the Bauer Core Facility at Harvard for infrastructure and support, the HMS Electron Microscopy Facility for EM sample preparation, C. Dulac and T. Hensch at Harvard University for providing cell culture infrastructure, J.C. Tapia for helpful discussion on the use of saponin, and A. Chamot for help with tissue section preparation. This work was supported by NIH grants U19 NS104653, UG3 MH123386, P50 MH094271 (J.W.L.), U24 NS109113 (J.S.T.), and K99 MH128891 (X.L.); NSF grants NSF-CAREER-2239688 (D.W.), NCS-FO-2124179, and NCS-FO-1835231 (H. Pfister.); Office of Naval Research grant N00014-20-1-2828 (J.W.L.). X. H. was supported by the Edward R. and Anne G. Lefler predoctoral fellowship from the Lefler Center for Neurodegenerative Disorders at Harvard Medical School and the Simmons Awards from the Harvard Center for Biological Imaging.

## Author contributions

X.H., X.L., J.W.L., and J.S.T. conceived this study. J.S.T. generated the mAb sequences. X.H. generated the scFv probes. T.F. and H. Ploegh provided materials for scFv probe production. X.H. performed the LM experiments. X.H. and R.S. performed the EM experiments. S.W. performed the imaging processing of the ssEM dataset. P.H.L. and V.J. performed 3D segmentation and curated the vCLEM dataset at Neuroglancer. Y.M. (yaron.mr@gmail.com) performed 2D segmentation. K.D.M. and X.H. performed IHC for scFv validation. L.M.M. and X.H. performed ICC for scFv validation. D.R.B. provided the reconstruction tool VAST and advice on 3D reconstruction and rendering. Y.W. helped with LM and EM co-registration. X.H., X.L., E.S.M., and S.A. performed 3D reconstruction and analysis. D.W., Z.L., J.A., and H. Pfister performed mitochondria and vesicle detection. X.H. and J.W.L. wrote the paper with input from J.S.T., H. Ploegh, R.S., D.R.B., Y.M., V.J., and P.H.L.

## Competing interests

The authors declare no competing interests.
