## [Peer Review File · Nature Communications]

Multiplexed volumetric CLEM enabled by scFvs provides new insights into the cytology of cerebellar cortexREVIEWER COMMENTS

Reviewer #1 (Remarks to the Author):

The authors developed eight smaller single-chain variable fragments (scFvs) based on eight well-characterized mAbs and conjugated them with various fluorescent dyes. This idea is great for volume correlative light and electron microscopy. The authors reported that each scFv proved effective as a detergent-free immunofluorescent probe. The approach was believed to be promising for routine linking of molecular information to connectomic information from the same material since the quality of data from the volumetric fluorescent and electron microscopy is good. However, some critical aspects of the experimental design need to be addressed, and additional experiments are required to draw conclusive findings.

1. One concern in this study is to choose Triton X-100 as a detergent for comparison. While Triton X-100 is commonly used to aid antibody penetration in light microscopy, it is not widely used for EM or immuno-EM studies. For EM studies, saponin is one of organic solvents which dissolve lipids from cell membranes making them permeable to antibodies. Furthermore, organic solvents can be used to fix and permeabilize cells at the same time by coagulating proteins. Saponin interacts with membrane cholesterol, selectively removing it and leaving holes in the membrane. Most researchers in the EM field choose low concentration of saponin to balance the antibody penetration and excellent membrane morphology in their immuno-EM experiments. Majority of their immuno-EM images showed great morphology of membrane at the ultrastructural level with the use of saponin. In some immuno-EM studies, Triton X-100 has been used. But those studies don't care the cell membrane, most of them were interested in some subcellular organelles.

2. Another point of concern is the lack of clarity on how the antibodies penetrate the cell through the cell membrane in detergent-free immunofluorescence labeling. A more thorough investigation and explanation of this process are needed to provide a comprehensive understanding of the technique's efficacy.

3. The animals were perfused with the fixative (4% paraformaldehyde + 0.1% glutaraldehyde). The low concentration of glutaraldehyde (0.1%) may not be sufficient to adequately preserve the lipids in the cell membrane. However, for detergent-free immunofluorescence labeling, scFv probes or nanobody probes were incubated for 3 days (50 μm) or 7 days (120 μm). This could potentially affect the ultrastructural morphology.

4. This study lack of novelty. The use of scFv has been well-established over the years, making it a solid foundation for the volume CLEM study. However, the choice of detergent-free immunofluorescence labeling raises questions about its suitability. Considering alternative approaches, such as utilizing low concentrations of saponin, might offer a more effective option for preserving membrane morphology during immuno-EM experiments.

5. The choice of 0.3% Triton X-100 to demonstrate detergent issues in EM seems excessive. Most EM studies recommend not exceeding 0.1% Triton X-100, making it unnecessary to use a higher concentration for this purpose.

Reviewer #2 (Remarks to the Author):

This is a technology development manuscript describing the generation and implementation of an assortment of single chain antibody-based probes (scFvs) against different brain proteins to label tissue for correlative fluorescence/volumetric electron microscopy. The key advance is the nature of the labeling probes, which can diffuse deep into fixed brain tissue and penetrate cells without the need for detergent permeabilization. The authors do nice side-by-side comparisons with whole IgG antibodies to highlight this. Detergent-free labeling thus allows processing for ultrastructural analysis

by EM, with excellent membrane preservation. Furthermore, the scFv labeling reagents can be easily labeled with different fluorescent dyes allowing the authors to visualize numerous (here they show 6) different labels in the same sample using spectral unmixing confocal microscopy. Serial section EM images of the same samples were then reconstructed and aligned with the fluorescence images to achieve correlated fluorescence/ultrastructure. Overall the data were compelling, with many beautiful examples of correlated fluorescence localization with 3D ultrastructure, nicely demonstrating the power of the technique. While the manuscript primarily focuses on tool development and offers little in the way of novel biological insight, I feel the potential future impact of the technique (i.e. ability to assign neural identities to volumetric connectomics EM datasets, ultrastructural localization of channels, receptors, etc.) will have broad appeal. There are several specific points that deserve attention:

-Nowhere in the manuscript do the authors validate their labeling reagents in a knockout background. In many cases the labeling pattern is distinct, consistent with previously published work and the localization of the signal makes sense with the correlated ultrastructure (i.e. vGLUT labels presynaptic terminals), but in some cases it is more ambiguous. For example, in Fig. S2a,e the authors argue that the CB and PV scFvs label more of the target proteins in the cell nucleus, is this real signal or are these probes picking up something non-specific in the nucleus that the IgG does not?

-While many of the images are quite striking, overall the manuscript lacked any sort of quantitative analysis. Just as one example, in Fig. 1, showing a simple pixel correlation scatter plot comparing the YFP and GFP-scFv signal would give readers a better idea of how evenly the scFv is penetrating cells to label YFP.

-p. 9 ".....immunofluorescence patterns that were similar to or in some cases stronger than their parental mAbs in Crus 1 of the cerebellar cortex (Figure 2 a; Sup. Figure 2; Sup. Figure 3). In many cases the comparisons between mAb and scFv is not entirely fair since the mAb is labeled in the 488/green channel (in which brain tissue notoriously has more autofluorescence) and the scFv in the red channel e.g. NPY signal in Sup. 2b,d; PSD95 in Sup. 2f. Is the labeling really that much cleaner or is the background signal in the green channel making the mAb appear worse than it is?

-p. 9 ".....found that the anti-calbindin scFv penetrated to a depth of ~150 μm in a 300- μm tissue slice". So the probe labeled throughout the entire slice?

**REVIEWER COMMENTS**

**Reviewer #1 (Remarks to the Author):**

*The authors developed eight smaller single-chain variable fragments (scFvs) based on eight*
*well-characterized mAbs and conjugated them with various fluorescent dyes. This idea is great*
*for volume correlative light and electron microscopy. The authors reported that each scFv*
*proved effective as a detergent-free immunofluorescent probe. The approach was believed to*
*be promising for routine linking of molecular information to connectomic information from the*
*same material since the quality of data from the volumetric fluorescent and electron microscopy*
*is good. However, some critical aspects of the experimental design need to be addressed, and*
*additional experiments are required to draw conclusive findings.*

*1. One concern in this study is to choose Triton X-100 as a detergent for comparison. While*
*Triton X-100 is commonly used to aid antibody penetration in light microscopy, it is not widely*
*used for EM or immuno-EM studies. For EM studies, saponin is one of organic solvents which*
*dissolve lipids from cell membranes making them permeable to antibodies. Furthermore,*
*organic solvents can be used to fix and permeabilize cells at the same time by coagulating*
*proteins. Saponin interacts with membrane cholesterol, selectively removing it and leaving holes*
*in the membrane. Most researchers in the EM field choose low concentration of saponin to*
*balance the antibody penetration and excellent membrane morphology in their immuno-EM*
*experiments. Majority of their immuno-EM images showed great morphology of membrane at*
*the ultrastructural level with the use of saponin. In some immuno-EM studies, Triton X-100 has*
*been used. But those studies don't care the cell membrane, most of them were interested in*
*some subcellular organelles.*

We thank the reviewer for pointing out the important fact that saponin is a far better
detergent for electron microscopy ultrastructural studies than Triton X-100. In response to this
suggestion, we did a new series of experiments analyzing twenty-two tissue blocks at various
saponin concentrations, sample thicknesses, and durations of antibody incubation. Moreover,
directly conjugated mAbs have become available, so we moved the results related to the
penetration tests with secondary antibody labeling (original Figure 1 d and e) to Supplementary
Figure 3 b and c). The new results are now presented in new Figure 1 panels f-i, new
Supplementary Figure 4 (plus additional new Supplementary Figure 5 that we will describe
below in point 4). The key result is that saponin at 0.05% concentration does not allow
fluorescently labeled monoclonal antibodies to penetrate into the middle 500 μm of a 1-mm
block even after a 1-week incubation with the labeled antibody (Figure 1 g; Supplementary
Figure 4). In contrast, the fluorescently labeled scFv penetrated throughout a block in the
absence of detergent (Figure 1 f; Supplementary Figure 4). In different experiments, we did
examine if higher concentrations of saponin could aid in deeper penetration (see new
Supplementary Figure 5) but for our purposes, even saponin at 0.05% was problematic. The
reason was that we found small breaks in the plasma membranes of neuronal processes
(arrows, Fig 1 i) that were not present in samples not treated with detergent (Fig 1 h). While
these ultrastructural breaks are small and, for many kinds of studies, would be of no
consequence, for connectomics they are serious. This seriousness is related to the requirement
for automatic algorithms to segment each nerve cell process. When two adjacent objects have a

continuity between them, this is often interpreted by the algorithms erroneously as the same
object. Such merge errors are far more difficult to find and correct than split errors, so avoiding
them at all costs is necessary (Shapson-Coe et al. 2021; Januszewski et al. 2018). With newer
techniques like multicolor 2-photon microscopy (Mahou et al. 2012; Blanc et al. 2023; Pudavar
et al. 2024), lightsheet microscopy in uncleared tissue (Schmid et al. 2013), and confocal done
with clearing approaches compatible with electron microscopy (Furuta et al. 2022), we think
having fluorescent scFv penetration hundreds of microns into tissue blocks will be of great use
in CLEM studies. We have modified the text to make these points clearer (line 150).

**Figure 1. Fluorescent scFv probes label brain tissues without detergents to preserve electron**
 **microscopy ultrastructure.**

**a**, Schematic representations of a full-length IgG antibody and an scFv probe with a conjugated
 fluorescent dye. **b**, Confocal images from the cerebral cortex of a YFP-H mouse labeled using a GFP-
 specific scFv probe conjugated with the red dye 5-TAMRA. Arrows show thinner neuronal processes,
 perhaps myelinated, that are not labeled by scFv. **c**, Layer 2/3 of the cerebral cortex labeled with a
 calbindin-specific scFv probe. **d**, Cerebellum cortex of Crus 1 labeled with the PSD-95 specific scFv. Right
 panel is the enlarged boxed inset from left. **e**, Cerebral cortex labeled with the NPY-specific scFv, **h**
 Tissue penetration comparison of a parvalbumin-specific scFv without detergent and its parental

mAbs directly conjugated with fluorophores with 0.05% saponin on 1-mm cerebral cortex tissue sections
with a 7-day incubation. **h** and **i**, Comparison of ultrastructure from samples incubated 7 days without
detergent and with 0.05% saponin. Arrows indicate membrane breaks. Asterisks indicate abnormal
appearing vesicle-filled axonal terminals.

**Sup. Figure 4. Penetration of anti-calbindin scFv into 1-mm tissue sample.**

Tissue penetration depth comparison of a calbindin-specific scFv without detergent and its parental mAbs
directly conjugated with fluorophores with 0.05% saponin on 1-mm cerebral cortex tissue sections with a
7-day incubation.

*2. Another point of concern is the lack of clarity on how the antibodies penetrate the cell through*
*the cell membrane in detergent-free immunofluorescence labeling. A more thorough*
*investigation and explanation of this process are needed to provide a comprehensive*
*understanding of the technique's efficacy.*

We agree with the reviewer on the importance of investigating how scFvs penetrate the
cell membrane in the absence of detergent. We are interested in determining the mechanism as
well. We think there are at least two possible mechanisms:

First, because all the immunolabeling experiments in this study were performed on brain
tissue samples from animals perfused and fixed with 4% formaldehyde (prepared fresh from
paraformaldehyde) + 0.1% glutaraldehyde in PBS, the cell membrane penetration by scFvs may
be simply explained by the fact that formaldehyde and glutaraldehyde permeate the lipid bilayer.
formaldehyde and glutaraldehyde are commonly used as chemical fixatives by crosslinking
amino groups of proteins (Fischer et al. 2008). It has been known that formaldehyde also

dissolves lipids (Fox et al. 1985; Kiernan 2000; Thavarajah et al. 2012). A recent study using
surface plasmon resonance (SPR) (Cheng et al. 2019) showed that fixation with formaldehyde
perturbed the integrity of membranes ($10 \pm 5\%$ mass loss), and they showed increased
permeability of sucrose. In another recent study using atomic force microscopy (Ichikawa et al.
2022), both formaldehyde and glutaraldehyde were shown to increase the size of nanoscopic
protrusions on cell membranes. These protrusions were generated by membrane protein
aggregates induced by crosslinking via formaldehyde or glutaraldehyde. The aggregated
membrane proteins may create gaps between them and their nearby lipids providing a
permeability pore. Additionally, two extracellular space-preserving fixation methods employing
formaldehyde and glutaraldehyde (Fulton and Briggman 2021; Lu et al. 2023) showed that full-
length antibodies can penetrate cell membranes albeit with lower diffusivity than scFvs,
supporting the idea that the formaldehyde plus glutaraldehyde treated membranes do have
gaps caused by the fixation.

We have tested this idea as well by using scFv immunolabeling on HEK293T cells
cultured as a single layer on a petri dish with a coverglass bottom. HEK293T cells allowed us to
avoid the issue of cut/fragmented cells in tissue sections where scFv could penetrate into cells
via a cut surface rather than through a membrane. After transfecting the HEK293T cells with a
plasmid encoding calbindin, we fixed the cells with the same fixative (4% formaldehyde + 0.1%
glutaraldehyde in PBS) for 15 min and then washed with PBS. Overnight immunolabeling of the
anti-calbindin scFv was then performed without or with 0.1% Triton-X. The results showed that
in both conditions (without or with 0.1% Triton-X), the scFv can penetrate and label its
intracellular target (we have added a new Supplementary Figure 7 a, b to show this data). This
result provides evidence consistent with the idea that the cell membranes fixed with 4%
formaldehyde + 0.1% glutaraldehyde allow scFvs to penetrate into intracellular spaces.
Additionally, we also tested a 1-hour immunolabeling protocol using scFvs and full-size mAbs
directed to transfected calbindin in COS-1 cells both without and with detergent
permeabilization. Similarly, we found that the scFvs were able to penetrate COS-1 cells and
label intracellular targets. However, the mAb was unable to penetrate at least at 1 hour (see
new Supplementary Figure 8). In another experiment we did find that an overnight incubation
with a mAb did label fixed cells that were not permeabilized with detergent. From all of these
experiments we infer that due to their small size the scFvs are better to penetrate fixed cells
than larger immunoprobos. We have modified the text to make these points clearer (line 169).

It is also possible that scFvs by virtue of their small size could permeate unfixed lipid
bilayers. Indeed (Li et al. 2016) showed that anti-pTau nanobodies when injected into the blood
of live mice could cross the blood-brain barrier and also cross neuronal cell membranes to label
intracellular pTau. In (Bernard et al. 2016), after transgenically inducing expression of an anti-
Otx2 scFv to express in cells of the choroid plexus cells, scFv in the CSF can cross the blood-
brain barrier and neutralize Otx2 in the cortex, perhaps via transcytosis. In (Thiel et al. 2002),
scFvs were shown to be able to pass through live cornea with an intact epithelium. (Im, Chung,
and Jang 2017) showed that scFvs can enter live, unfixed culture cells.

Based on these results, we were motivated to see if the scFvs we generated could cross
into living cells. We attempted to immunolabel the transfected HEK293T cell for calbindin with
the anti-calbindin scFv using live HEK293T cells. We found that after a one-hour incubation, the
scFv could penetrate cells (Supplementary Figure 7 c, arrow). However, unlike the penetration
of fixed cells described above, the labeling was more punctate. This labeling was most likely

explained by endocytosis as has been previously seen for extracellular dye molecules (see for
example, (Tsuriei et al. 2015)). We have added a Supplementary Figure 7 c to show this result.
Consistent with this, it has been documented that both nanobodies and scFvs can be
internalized into cells via endocytosis (de Beer and Giepmans 2020; Wittrup et al. 2009; Alric et
al. 2018; Kim et al. 2020). We have modified the text to make this point clearer (line 176). This
is a potentially important route of entry because it provides an option to achieve immunolabeling
of larger tissue samples, such as a whole mouse brain, by introducing these small immuno-
probes in live animals.

Sup. Figure 7. Penetration of anti-calbindin scFv into fixed HEK cells or live cells.

Immunofluorescence immunocytochemistry on transiently transfected cells. HEK cells were transfected
with a plasmid encoding Flag-tagged human calbindin. **a**, Chemically fixed cells were labeled overnight
with Alexa594 anti-calbindin L109/57 scFv in the absence of detergent. **b**, Chemically fixed cells were
labeled overnight with Alexa594 anti-calbindin L109/57 scFv with 0.1% Triton-X. **c**, Live cells were labeled
with Alexa594 anti-calbindin L109/57 scFv. The arrow indicates the cell that has intracellular scFv
labeling. Arrowheads indicate puncta labeling in some cells.

**Sup. Figure 8. Penetration of anti-calbindin scFv into fixed COS-1 cells.**

Immunofluorescence immunocytochemistry on transiently transfected cells. COS-1 cells were transfected
with a plasmid encoding Flag-tagged human calbindin. Cells in panels A and B were labeled for 1 hour
after fixation and prior to detergent permeabilization with (A) Alexa594 anti-calbindin L109/57 scFv or (B)
anti-calbindin mouse mAb L109/39 (scFv and mAb labeling in red). After permeabilization, cells were
labeled with rabbit anti-Flag (green) to detect calbindin, and Hoechst nuclear dye (blue). For cells in
panels C and D all immunolabeling was performed after fixation and detergent permeabilization with (C)
Alexa594 anti-calbindin L109/57 scFv or (D) anti-calbindin mouse mAb L109/39 (scFv and mAb labeling
in red). Cells were simultaneously labeled with rabbit anti-Flag (green), and Hoechst (blue). Cells in all
panels were imaged at the same exposure.

*3. The animals were perfused with the fixative (4% paraformaldehyde + 0.1% glutaraldehyde).*
*The low concentration of glutaraldehyde (0.1%) may not be sufficient to adequately preserve the*
*lipids in the cell membrane. However, for detergent-free immunofluorescence labeling, scFv*
*probes or nanobody probes were incubated for 3 days (50 μm) or 7 days (120 μm). This could*
*potentially affect the ultrastructural morphology.*

We agree this is a reasonable concern that the use of 0.1% glutaraldehyde does not
sufficiently preserve lipids in the cell membrane, which may cause the ultrastructure to
deteriorate when tissue samples are incubated with immuno-probes for prolonged periods like

three or seven days. We were aware of this potential problem. The reasons we used 0.1%
glutaraldehyde instead of a higher concentration was: first, glutaraldehyde is a harsher fixative
which may modify epitopes on target proteins (Fischer et al. 2008), preventing immuno-probe
labeling; Second, glutaraldehyde has higher autofluorescence than formaldehyde (Fischer et al.
2008), which causes high background in fluorescence microscopy. Because we only used 0.1%
glutaraldehyde, we always postfixed the perfused brain for many hours (overnight). To prevent
reversal of the formaldehyde fixation (Fischer et al. 2008) the brain samples were then sliced in
ice-cold fixative (4% formaldehyde + 0.1% glutaraldehyde) and stored in the same fixative at
4 °C. The only exception to this protocol was our work with the neuropeptide NPY, which we, as
others, have found to be difficult to label if the fixation is too extensive. In this case, we stored
the slices in PBS at 4 °C. We also performed all the incubations, including the washing steps, at
4 °C to prevent ultrastructural degradation. In the manuscript, in Supplementary Figure 21, we
examined the ultrastructure of a 2 mm, 2 mm, 120- μ m thick cerebellum tissue sample incubated
with scFv probes for seven days after light fixation (described above). As shown in the figure,
ultrastructure at four locations across the cerebellar cortex layers including regions that are near
the center of the block, is preserved well. After careful examination of the images during the
revision process, we have noticed some abnormalities. In the superficial layer (the molecular
layer) of the cerebellar cortex, which is mainly composed of neuronal processes and close to
the surface of the block, we did observe some artifacts (new arrows in Supplementary Figure
21). We are unsure whether these artifacts are explained by mechanical or chemical or thermal
factors that are different at the surface vs. the interior of the block. We have also modified the
manuscript (line 217) to make readers aware of this issue. If reviewers are interested in
examining the ultrastructure directly, we encourage reviewers to visit the Neuroglancer link of
our vCLEM dataset at Neuroglancer LINK.

In addition, in a new set of experiments we performed for the revision that we will
discuss in detail below in point 4, we showed that instead of 3-day or 7-day incubations, the
anti-calbindin scFv can penetrate to the center of a 300- μ m vibratome section with incubation of
only one day. We found fewer tissue artifacts in the ultrastructure of 1-day incubated samples
than the 7-day samples (see an example in new Supplementary Figure 6, arrows indicating
artifacts). We have modified the text to make this point clearer (line 166). So, we conclude that,
at least for some scFvs, 1-day incubations are sufficient.

Sup. Figure 6. Ultrastructure comparison between samples incubated for one day or seven days.

Ultrastructure of locations close to the surfaces of 300- μm cerebral cortex sections immunolabeled for
one day (**a**) or seven days (**b**). Arrows indicate artifacts.

**Sup. Figure 21. Well-preserved ultrastructure from the surface (a) to the middle (d) of the 120- μ m**
 **section.**

Panel 1-4 in a-d show the ultrastructure at the locations labeled by the red circles in the right panels.
 Arrows indicate the artifacts potentially caused by prolonged incubation with scFvs for immunolabeling.

*4. This study lack of novelty. The use of scFv has been well-established over the years, making*
 *it a solid foundation for the volume CLEM study. However, the choice of detergent-free*
 *immunofluorescence labeling raises questions about its suitability. Considering alternative*
 *approaches, such as utilizing low concentrations of saponin, might offer a more effective option*
 *for preserving membrane morphology during immuno-EM experiments.*

We agree with the reviewer that the use of scFvs is well-established (Bird et al. 1988;
 Huston et al. 1988; Monnier, Vigouroux, and Tassew 2013; Ahmad et al. 2012). The use of
 scFvs as immuno-probes for CLEM has been raised in a number of papers in discussion (de
 Beer and Giepmans 2020; Franek et al. 2024) but, to the best of our knowledge, this is the first
 actual demonstration of scFvs in volumetric CLEM.

We agree with the reviewer that when performing volumetric CLEM, alternative
 immunolabeling approaches other than those that employ scFvs should be considered, such as

fluorescently tagged primary IgG antibodies with saponin permeabilization. Therefore, in new
experiments we compared detergent-free immunolabeling with scFv and Triton-X or saponin-
enabled immunolabeling with a dye-directly conjugated monoclonal antibody (mAb) at various
detergent concentrations (0.1% and 0.3% Triton-X; 0.05%, 0.1%, and 0.2% saponin) and with
two different incubation times (1 day and 7 days). The experiments were performed on 300- μ m
cerebral cortex tissue blocks with an anti-calbindin L109/57 scFv and a dye-directly conjugated
anti-calbindin L109/57 mAb. The epitope binding site of the mAb and the scFv were the same.
As shown in new Supplementary Figure 5 a, after 1 day of incubation, only the scFv and the
mAb with 0.3% Triton-X penetrated to the middle (i.e., 150 μ m) of the section. The other
experimental conditions showed various degrees of penetration: 0.1% Triton-X, \sim 50 μ m; 0.05%
saponin, \sim 30 μ m; 0.1% saponin, \sim 80 μ m; 0.2% saponin, \sim 100 μ m. In all cases with saponin
permeabilization, there was a lack of labeling in the cell nuclei (indicated by arrows). When we
examined the ultrastructure of these labeled samples, the samples treated with detergent-free
scFv labeling showed the best quality. The sample treated with 0.05% saponin showed good-
quality EM ultrastructure. All the other samples showed compromised EM ultrastructure, the
severity of which increased with the increase of detergent concentration. The membrane breaks
in these samples would make automatic segmentation for connectomics challenging, as stated
above in our answer to point 1. Although the sample treated with 0.05% saponin for one day
showed no obvious ultrastructural artifacts, the mAb penetration was far shallower than the scFv
(\sim 30 μ m vs. 150 μ m) making volumetric CLEM on the samples larger than the penetration depth
difficult.

As shown in new Supplementary Figure 5 b, after 7 days of incubation, scFvs without
detergent and mAb with various concentrations of Triton-X or saponin can penetrate the middle
of the 300- μ m. However, we still observed in the case of 0.05% saponin a lack of labeling in the
cell nuclei (indicated by arrows). Again, when examining the ultrastructure of these labeled
samples, the sample treated with detergent-free scFv labeling showed the best quality and is
similar to the one-day sample (which we have also mentioned in our answer to reviewer's point
3). All the other samples showed compromised EM ultrastructure, which was much worse when
compared with the 1-day samples. Even the 0.05% saponin now showed membrane breaks.
We also noticed after 7-day saponin incubation a new artifact: the vesicle-filled axonal profiles in
samples treated with saponin for seven days showed a granular texture (indicated by
arrowheads in Supplementary Figure 5 b). We think the protein-coagulating function of saponin,
as the reviewer stated previously, may be the cause. These granules could pose challenges
when synaptic vesicles need to be automatically detected and analyzed (as we did in the later
part of this paper) for connectomic studies.

In addition, as we have mentioned in our answer to point 1, scFvs can penetrate 1-mm
tissue blocks while saponin at 0.05% concentration only allows mAbs to penetrate into 250 μ m
after a seven-day incubation (Figure 1 g; new Supplementary Figure 4). These new results
suggest that if a researcher wants to do a small-scale volumetric CLEM on a smaller tissue
sample (such as several μ m to 50- μ m), directly dye-conjugated primary antibodies with a low
concentration (0.05%) of saponin with a shorter incubation (one day) may be an option.
However, should a researcher need to conduct large-scale volumetric CLEM on larger tissue
samples (\sim 1 mm in thickness), using scFvs for detergent-free immunolabeling is more
advantageous. Large-scale volumetric CLEM is especially important for connectomics because
a smaller volume is very likely to have fragmented cells/processes that prevent the mapping of
the neural circuits. We have modified the text to make these points clearer (line 150; line 162).

Sup. Figure 5. Tissue penetration depth comparison of scFvs in the absence of detergent and fluorophore-conjugated mAbs with the treatments of various concentrations of detergents.

300- μm cerebral cortex sections were immunolabeled for one day (a) or seven days (b) with a calbindin-
specific scFv conjugated with 5-TAMRA in the absence of detergent or with the scFv's parental mAb
conjugated with FL550 in the presence of 0.1%, 0.3% Triton-X, or 0.05%, 0.1%, 0.2% saponin. Arrows
indicate unlabeled cell nuclei. Arrowheads indicate granular textures associated with the treatment of
saponin.

*5. The choice of 0.3% Triton X-100 to demonstrate detergent issues in EM seems excessive.*
*Most EM studies recommend not exceeding 0.1% Triton X-100, making it unnecessary to use a*
*higher concentration for this purpose.*

We agree with the reviewer that choosing 0.3% Triton X-100 is excessive to
demonstrate the detergent's issue on EM ultrastructure. We have changed Figure 1 h and i so
the comparison is with a sample treated with 0.05% saponin.

**Reviewer #2 (Remarks to the Author):**

*This is a technology development manuscript describing the generation and implementation of*
*an assortment of single chain antibody-based probes (scFvs) against different brain proteins to*
*label tissue for correlative fluorescence/volumetric electron microscopy. The key advance is the*
*nature of the labeling probes, which can diffuse deep into fixed brain tissue and penetrate cells*
*without the need for detergent permeabilization. The authors do nice side-by-side comparisons*
*with whole IgG antibodies to highlight this. Detergent-free labeling thus allows processing for*
*ultrastructural analysis by EM, with excellent membrane preservation. Furthermore, the scFv*
*labeling reagents can be easily labeled with different fluorescent dyes allowing the authors to*
*visualize numerous (here they show 6) different labels in the same sample using spectral*
*unmixing confocal microscopy. Serial section EM images of the same samples were then*
*reconstructed and aligned with the fluorescence images to achieve correlated*
*fluorescence/ultrastructure. Overall the data were compelling, with many beautiful examples of*
*correlated fluorescence localization with 3D ultrastructure, nicely demonstrating the power of the*
*technique. While the manuscript primarily focuses on tool development and offers little in the*
*way of novel biological insight, I feel the potential future impact of the technique (i.e. ability to*
*assign neural identities to volumetric connectomics EM datasets, ultrastructural localization of*
*channels, receptors, etc.) will have broad appeal. There are several specific points that deserve*
*attention:*

*-Nowhere in the manuscript do the authors validate their labeling reagents in a knockout*
*background. In many cases the labeling pattern is distinct, consistent with previously published*
*work and the localization of the signal makes sense with the correlated ultrastructure (i.e.*
*vGLUT labels presynaptic terminals), but in some cases it is more ambiguous. For example, in*
*Fig. S2a,e the authors argue that the CB and PV scFvs label more of the target proteins in the*
*cell nucleus, is this real signal or are these probes picking up something non-specific in the*
*nucleus that the IgG does not?*

Concerning validation, we agree with the reviewer that the most crucial concern for
immuno-probes or any similar affinity probes is whether they label or detect the actual target
they are supposed to bind to. There are many cases when antibodies working in ELISA or
Western blot settings fail to label their targets or have off-target labeling that creates abnormal
background signals in immunohistochemistry (IHC). The parental (aka. progenitor) monoclonal
antibodies (mAbs) from the UC Davis/NIH NeuroMab facility, whose sequences were used to
generate the eight scFvs in this study, have undergone a strict validation process. In all but one
case (the anti-NPY mAb), the mAbs have passed by at least three of the following:
immunofluorescence on transfected COS-1 cells, Western blots on homogenized rat and mouse
brains, IHC on rat and mouse brain sections, and IHC on mouse sections in a knockout
background. These were accomplished in co-author James Trimmer's lab (for more details, see
(Gong, Murray, and Trimmer 2016)). The validation tests of the eight mAbs used in the paper
are now shown in new Supplementary Table 4). Although limited by the availability of KO brain
samples, the three that we were able to test of them (N206Bb/9, GFAP R416WT; K28/43, PSD-
95; K14/16, Kv 1.2) have passed the test of IHC on WT versus KO mouse brain sections, in that
all detectable labeling observed in WT sections was eliminated in KO sections (all three also
passed on WT/KO comparison by immunoblot) in a knockout background. While we want to test
all the mAbs in a knockout background, but we hope the reviewer understands that it is

challenging to gather KO animals brain samples for all seven endogenous targets because
 some may be lethal mutations.

We also validated the scFvs in each case via IHC on rat and mouse brain sections and
 by immunofluorescence immunocytochemistry on transiently transfected COS-1 cells (also
 summarized in Supplementary Table 4; we also provide representative images in new
 Supplementary Figure 9, 10, 11 for the validation of the N206b/9, anti-GFAP R416WT scFv.
 Details of the methods of the validation tests for the other scFvs in this paper (and other scFvs)
 can be found in (Mitchell et al. 2023; Gong, Murray, and Trimmer 2016). We have modified the
 text to make these points clearer (line 139;line 143).

**Sup. Table 4. Information on the validation of the scFvs and their parental mAbs.**

Target	Clone No.	mAb validation				scFv validation
		COS-IF	Brain IB	Brain IHC	KO Brain IHC	Method
GFP	N86/38	Pass	NA	NA	NA	COS-IF
Calbindin	L109/57	Pass	Pass	Pass	ND	Brain IHC and COS-IF
GFAP R416WT	N206B/9	Pass	Pass	Pass	Pass	Brain IHC and COS-IF
VGluT1	N28/9	Pass	Pass	Pass	ND	Brain IHC and COS-IF
PSD-95	K28/43	Pass	Pass	Pass	Pass	Brain IHC and COS-IF
Kv 1.2	K14/16	Pass	Pass	Pass	Pass	Brain IHC and COS-IF
Parvalbumin	L114/81	Pass	Pass	Pass	ND	Brain IHC and COS-IF
NPY	L115/13	Pass	Fail	Pass	ND	COS-IF

Sup. Figure 9. Validation of anti-GFAP scFv with Immunofluorescence immunocytochemistry.

Immunofluorescence immunocytochemistry on transiently transfected cells. COS-1 cells (top row) and
 HEK293T cells (bottom row). Cells were transfected with a plasmid mEmerald-tagged human GFAP

(green) and double immunolabeled with the 5-TAMRA-labeled anti-GFAP N206B/9 scFv (red) and the
anti-GFAP N206A/8 mouse IgG1 mAb (white). Hoechst nuclear labeling is shown in blue.

**Sup. Figure 10. Validation of anti-GFAP scFv with Immunofluorescence immunohistochemistry**
**(cerebellum).**

GFAP scFv and original monoclonal antibody from which it was derived display the same tissue labeling
pattern of a sagittal section through the rat cerebellum. **A)** Glial cells throughout the cerebellar granule
cell layer (GCL) and prominent Bergmann glial process in molecular layer (ML) are labeled with
hybridoma derived monoclonal antibody N206B/9. **B)** merged image includes labeling with a polyclonal
rabbit antibody (KC) against the neuronal potassium channel Kv2.1, monoclonal antibody targeting glial
specific RNA binding protein QKI (N147/6) and nuclear specific Hoechst labeling. **C)** An adjacent section
labeled with N206B/9 derived scFv shows the same pattern of labeling. **D)** merged image with the same
additional labeling as B.

**Sup. Figure 11. Validation of anti-GFAP scFv with Immunofluorescence immunohistochemistry**
 **(hippocampus).**

Validation of scFv labeling pattern against hybridoma-generated monoclonal antibody N206B/9 from
 which it was derived. Multiplex immunofluorescent labeling of a sagittal section through rat hippocampal
 region CA1. **A)** 5-TAMRA conjugated N206B/9 derived scFv, **B)** Hybridoma derived monoclonal antibody
 N206B/9 indirectly labeled with Alexa fluor 647 conjugated goat anti-mouse IgG1 secondary antibody, **C)**
 merged images from A and B illustrating co-labeled astroglial cells (e.g arrowheads). **D)** Same multiplex
 image shown in C with additional neuronal specific potassium channel Kv2.1, glial specific pan-QKI RNA
 binding protein, and DNA marker Hoechst 33342 labeling.

The second issue raised is concern over the ambiguous signals detected by the anti-
 calbindin and anti-parvalbumin scFvs in the cell nuclei of Purkinje cells. Clearly, in the original
 Supplementary Figure 2, the mAbs for calbindin and parvalbumin did not detect signals in the
 cell nuclei as the scFvs did. In the revised paper we now include the validation IHC images of
 the anti-calbindin and anti-parvalbumin mAbs from experiments done in James Trimmer's lab
 (new Supplementary Figure 14). These images showed the normal expected labeling, that
 included signal in some cell nuclei of Purkinje cells in the lateral hemisphere of the rat
 cerebellum. In the original manuscript, we mentioned previous studies reporting the detection of
 calbindin and parvalbumin in cell nuclei of Purkinje cells (Celio 1990; Brandenburg et al. 2021;
 German et al. 1997; Schmidt et al. 2007). We believe calbindin and parvalbumin are present in
 the cell nuclei. We therefore suspect that the reason our mAb labeling didn't detect signals in
 the cell nuclei is insufficient antibody incubation time.

To address this problem, in Triton-X-treated samples, we first did immunolabeling with
the same anti-calbindin and anti-parvalbumin mAbs directly conjugated with the red fluorophore
FL-550 in distinction to using secondary antibodies as we did previously. The directly
conjugated mAbs exclude the requirement for secondary antibodies allowing for only one round
of incubation. We extended the incubation time to seven days (versus 2 days of incubation with
primary antibodies previously). The results showed that both mAbs can detect signals in the cell
nuclei of most (>~90%) Purkinje cells (new Supplementary Figure 15 c, l). This staining is
similar to the scFv labeling except that scFvs were detected in all Purkinje cells. The second
approach we attempted was immunolabeling with commercial polyclonal antibodies (pAbs)
against calbindin and parvalbumin. These antibodies are not directly conjugated with
fluorophores, so we utilized (Fab)₂ as fluorescently tagged secondaries, which are smaller than
conventional secondaries and supposedly can diffuse in tissue better. Our results showed that,
again, both calbindin and parvalbumin could be detected by the pAbs in the cell nuclei in most
but not all Purkinje cells (new Supplementary Figure 15 d and m). The third approach we used,
was to slice the section in a different orientation to cut through the nuclei of most Purkinje cells
in order to gain direct access to the nuclei in the vibratome section (see new Supplementary
Figure 15 e). We immunolabeled with the same anti-calbindin or anti-parvalbumin mAbs directly
conjugated with the fluorophore FL-550 with a seven-day incubation. This time, we observed
labeling in nearly all Purkinje cells (Supplementary Figure 15 f and n). These results indicate
that the signals detected in the cell nuclei by scFvs are real signals. The reason it is relatively
harder for full-length antibodies (mAbs or pAb) to detect these signals can be attributed to their
relatively weaker penetration ability, even in the presence of Triton-X. This can be addressed in
ways like extending incubation time or slicing sections to expose the internal epitopes in cell
nuclei better. We have modified the text to make these points clearer (line 148).

**Sup. Figure 14. Validation of anti-calbindin and anti-parvalbumin scFvs with Immunofluorescence**
 **immunohistochemistry.**

Labeling pattern of original mAbs used to generate scFvs against Parvalbumin and Calbindin in rat
 cerebellum. Sagittal section through cerebellum labeled with monoclonal antibodies L127/8 (A, GAD1),
 L114/8 R (B, PARV), and L109/57 (C, CALB1). The merged image (D) shows the colocalized pattern of
 labeling within Purkinje cell layer (PCL). ML, molecular layer, GCL, granule cell layer.

**Sup. Figure 15. Validation of immunofluorescence by scFv probes and their parental mAbs (part**
**3).**

Cerebellum Crus 1 sections were immunolabeled with a calbindin-specific scFv (a), or its parental mAb
and secondary antibody conjugated with Alexa Fluor 488 (b), the mAb conjugated with FL550 (c), or a
commercial calbindin-specific pAb and secondary (Fab)₂ conjugated with Alexa Fluor 594 (d). e,
Schematics showing the cutting orientation that is parallel to the lobule of Crus 1, which intersects
perpendicular to the planer Purkinje cells in Crus 1. f, Sections cut in this orientation immunolabeled with
the mAb conjugated with FL550. The boxed inset is shown enlarged in the adjacent panel. Whole-section
images of cerebellum Crus 1 sections immunolabeled with a calbindin-specific scFv (g), or its parental
mAb and secondary antibody conjugated with Alexa Fluor 488 (h), or the mAb conjugated with FL550 (i).
Arrows indicate labeled cell nuclei of Purkinje cells. Arrowheads indicate the labeled axons.

Cerebellum Crus 1 sections were immunolabeled with a parvalbumin-specific scFv (j), or its parental mAb
and secondary antibody conjugated with Alexa Fluor 488 (k), the mAb conjugated with FL550 (l), or a
commercial parvalbumin-specific pAb and secondary (Fab)₂ conjugated with Alexa Fluor 594 (m). f,
Sections cut in this orientation in e immunolabeled with the mAb conjugated with FL550. The boxed inset
is shown enlarged in the adjacent panel. Whole-section images of cerebellum Crus 1 sections
immunolabeled with a parvalbumin-specific scFv (o), or its parental mAb and secondary antibody
conjugated with Alexa Fluor 488 (p), or the mAb conjugated with FL550 (q). Arrows indicate labeled cell
nuclei of Purkinje cells. Arrowheads indicate the labeled axons.

*-While many of the images are quite striking, overall the manuscript lacked any sort of*
*quantitative analysis. Just as one example, in Fig. 1, showing a simple pixel correlation scatter*
*plot comparing the YFP and GFP-scFv signal would give readers a better idea of how evenly*
*the scFv is penetrating cells to label YFP.*

We thank the reviewer for highlighting the lack of quantitative analysis in comparing the
specificity of scFvs and mAbs. The paper does contain other quantitative analyses (see Figure
7; Supplementary Figure 29; Supplementary Table 5, 6) but in response to the specific question
raised, we have now created pixel correlation scatter plots for three images from two cortical
and one hippocampal section from YFP-H mice, which were also immunolabeled with the anti-
GFP scFv (new Supplementary Figure 2). Supplementary Figure 2 a is the raw image of Figure
1 b. As is shown in all three pixel correlating scatter plots, the signals from the scFv labeling
(red) correlate with the native YFP fluorescence signal (green). There are pixels that only have
values in the green channels, which correspond to the insufficiently labeled axons pointed out in
Figure 2 a. There are very few pixels that only have values in the red (scFv) channel, which
indicates that there is minimal off-target labeling of this anti-GFP scFv. This analysis gives us
confidence in the specificity of the scFv for green fluorescent protein. Doing this kind of double
labeling is more problematic when comparing scFvs to mAbs that have the identical paratope as
they compete for the same site. So, in these cases, as described in detail above, we had to be
content with the comparative labeling of different tissue sections. We have modified the text to
make this point clearer (line 128).

Sup. Figure 2. Pixel correlation scatter plots comparing the native YFP fluorescence signal and the red fluorescence from the labeling of the GFP- specific scFv.

Cerebral cortex samples (**a** and **b**) and hippocampus (**c**) from YFP-H mice were immunolabeled with a
GFP-specific scFv conjugated with the red fluorophore 5-TAMRA. The images are raw data without any
brightness/contrast adjustment. **a** is the raw image data of Figure 1 b.

*-p. 9 ".....immunofluorescence patterns that were similar to or in some cases stronger than their*
*parental mAbs in Crus 1 of the cerebellar cortex (Figure 2 a; Sup. Figure 2; Sup. Figure 3). In*
*many cases the comparisons between mAb and scFv is not entirely fair since the mAb is*
*labeled in the 488/green channel (in which brain tissue notoriously has more autofluorescence)*
*and the scFv in the red channel e.g. NPY signal in Sup. 2b,d; PSD95 in Sup. 2f. Is the labeling*

*really that much cleaner or is the background signal in the green channel making the mAb*
*appear worse than it is?*

We apologize that the phrasing in our original manuscript may have caused a
misunderstanding. In this sentence, “in some cases stronger than their parental mAbs in Crus 1
of the cerebellar cortex.” only refers to the cases of calbindin and parvalbumin, as discussed in
the previous point. In the other cases (GFAP, VGluT1, Kv 1.2, PSD-95, and NPY), we believe
our results suggest that the labeling of scFvs and mAbs are comparably good in terms of both
signal and background. We understand the legitimate concern that the tissue sections fixed with
formaldehyde and glutaraldehyde may have a higher background in the 488/green channel.
Glutaraldehyde especially has stronger autofluorescence (Fischer et al. 2008). However, we
only used 0.1% glutaraldehyde in our preparation prior to osmium staining. After adequate
washing with PBS, the brain tissue sections do not show strong autofluorescence in the
488/green channel (as now shown in new Supplementary Figure 12 a). There is some
autofluorescence, mostly from lipofuscin granules in cell bodies (arrows in new Supplementary
Figure 12 a) with broad excitation/emission spectra (Di Guardo 2015; Marmorstein et al. 2002).
But this autofluorescence is found in all channels. We emphasize that we are not attempting to
make a case that scFv labeling is cleaner than that obtained with mAb, as the results from both
are very similar. Indeed we also performed new experiments with red fluorophore-conjugated
mAbs for calbindin and parvalbumin, as discussed in our answer to the reviewer’s point 1, and
didn’t find any difference in the background level (new Supplementary Figure 15). We have
adjusted the phrasing in the manuscript (line 145) to avoid any further misunderstanding.

Sup. Figure 12. Validation of immunofluorescence by scFv probes and their parental mAbs (part 1).

**a**, Confocal images from unlabeled the cerebral cortex and cerebellar cortex of a wild-type mouse
showed limited background in the 488/green channel. Arrows indicate background signals lipofuscin
granule. **b-d**, Cerebellum Crus 1 sections were immunolabeled with scFvs targeting VGluT1, GFAP, and

Kv 1.2; or these scFvs' parental mAbs and secondary antibodies conjugated with Alexa Fluor 488.

*-p. 9 ".....found that the anti-calbindin scFv penetrated to a depth of ~150 μm in a 300-μm tissue*
*slice". So the probe labeled throughout the entire slice?*

We apologize for the lack of clarity, yes, we meant that they labeled through the entire
300 μm slice (150 μm from each side). Given the recent availability of directly conjugated mAbs
we have removed Figure 1 d and e to Supplementary Figure 3 b (we remade figures from raw
images showing the penetration across the 300 μm thickness) and c, and added new Figure 1 f-
i, Supplementary Figure 4 and 5 of the results of a comparable experiment of scFv labeling on
300-μm and 1-mm thick brain tissue sections in the absence of detergent. In Figure 1 and
Supplementary Figure 4, scFvs are shown to label throughout a 1-mm thickness with a seven-
549 day incubation; in Supplementary Figure 5, scFvs can label a 300-μm thickness sample with
550 either a 1- or 3-day incubation. EM ultrastructure of all these samples was good. We have
551 modified the text to make these points clearer (line 150).

**Figure 1. Fluorescent scFv probes label brain tissues without detergents to preserve electron**
 **microscopy ultrastructure.**

**a**, Schematic representations of a full-length IgG antibody and an scFv probe with a conjugated
 fluorescent dye. **b**, Confocal images from the cerebral cortex of a YFP-H mouse labeled using a GFP-
 specific scFv probe conjugated with the red dye 5-TAMRA. Arrows show thinner neuronal processes,
 perhaps myelinated, that are not labeled by scFv. **c**, Layer 2/3 of the cerebral cortex labeled with a
 calbindin-specific scFv probe. **d**, Cerebellum cortex of Crus 1 labeled with the PSD-95 specific scFv. Right
 panel is the enlarged boxed inset from left. **e**, Cerebral cortex labeled with the NPY-specific scFv. **f** and **g**,
 Tissue penetration comparison of a parvalbumin-specific scFv without detergent and its parental

mAbs directly conjugated with fluorophores with 0.05% saponin on 1-mm cerebral cortex tissue sections
 with a 7-day incubation. **h** and **i**, Comparison of ultrastructure from samples incubated 7 days without
 detergent and with 0.05% saponin. Arrows indicate membrane breaks. Asterisks indicate abnormal
 appearing vesicle-filled axonal terminals.

**Sup. Figure 3. Immunolabeling results of anti-calbindin scFv and tissue penetration comparison.**

**a**, Additional brain regions labeled with a calbindin-specific scFv probe conjugated with 5-TAMRA. The
 arrow in the left panel shows myelinated Purkinje cell axons in the granule layer. **b**, Tissue penetration

depth comparison of scFvs, mAbs (plus secondary antibodies), and the role of detergents in improving
the depth of labeling. **c**, Comparison of ultrastructure with and without 0.5% Triton X-100 on scFv labeled
samples. Boxed insets are shown at higher magnification in adjacent panels. of ~30 μm ; the nanobody
can penetrate into a depth of ~150 μm .

Sup. Figure 4. Penetration of anti-calbindin scFv into 1-mm tissue sample.

Tissue penetration depth comparison of a calbindin-specific scFv without detergent and its parental mAbs
directly conjugated with fluorophores with 0.05% saponin on 1-mm cerebral cortex tissue sections with a
7-day incubation.

Sup. Figure 5. Tissue penetration depth comparison of scFvs in the absence of detergent and fluorophore-conjugated mAbs with the treatments of various concentrations of detergents.

300- μm cerebral cortex sections were immunolabeled for one day (a) or seven days (b) with a calbindin-
specific scFv conjugated with 5-TAMRA in the absence of detergent or with the scFv's parental mAb
conjugated with FL550 in the presence of 0.1%, 0.3% Triton-X, or 0.05%, 0.1%, 0.2% saponin. Arrows
indicate unlabeled cell nuclei. Arrowheads indicate granular textures associated with the treatment of
saponin.

**References**

- Ahmad, Zuhaida Asra, Swee Keong Yeap, Abdul Manaf Ali, Wan Yong Ho, Noorjahan Banu
Mohamed Alitheen, and Muhajir Hamid. 2012. "scFv Antibody: Principles and Clinical
Application." *Clinical & Developmental Immunology* 2012 (March): 980250.
- Alric, Christophe, Katel Hervé-Aubert, Nicolas Aubrey, Souad Melouk, Laurie Lajoie, William
Mème, Sandra Mème, et al. 2018. "Targeting HER2-Breast Tumors with scFv-Decorated
Bimodal Nanoprobos." *Journal of Nanobiotechnology* 16 (1): 18.
- Beer, Marit A. de, and Ben N. G. Giepmans. 2020. "Nanobody-Based Probes for Subcellular
Protein Identification and Visualization." *Frontiers in Cellular Neuroscience* 14 (November):
573278.
- Bernard, Clémence, Clémentine Vincent, Damien Testa, Eva Bertini, Jérôme Ribot, Ariel A. Di
Nardo, Michel Volovitch, and Alain Prochiantz. 2016. "A Mouse Model for Conditional
Secretion of Specific Single-Chain Antibodies Provides Genetic Evidence for Regulation of
Cortical Plasticity by a Non-Cell Autonomous Homeoprotein Transcription Factor." *PLoS*
*Genetics* 12 (5): e1006035.
- Bird, R. E., K. D. Hardman, J. W. Jacobson, S. Johnson, B. M. Kaufman, S. M. Lee, T. Lee, S.
H. Pope, G. S. Riordan, and M. Whitlow. 1988. "Single-Chain Antigen-Binding Proteins."
*Science* 242 (4877): 423–26.
- Blanc, Hugo, Gabriel Kaddour, Nicolas B. David, Willy Supatto, Jean Livet, Emmanuel
Beaurepaire, and Pierre Mahou. 2023. "Chromatically Corrected Multicolor Multiphoton
Microscopy." *ACS Photonics* 10 (12): 4104–11.
- Brandenburg, Cheryl, Lindsey A. Smith, Michaela B. C. Kilander, Morgan S. Bridi, Yu-Chih Lin,
Shiyong Huang, and Gene J. Blatt. 2021. "Parvalbumin Subtypes of Cerebellar Purkinje
Cells Contribute to Differential Intrinsic Firing Properties." *Molecular and Cellular*
*Neurosciences* 115 (September): 103650.
- Celio, M. R. 1990. "Calbindin D-28k and Parvalbumin in the Rat Nervous System."
*Neuroscience* 35 (2): 375–475.
- Cheng, Ruoyu, Feng Zhang, Meng Li, Xiang Wo, Yu-Wen Su, and Wei Wang. 2019. "Influence
of Fixation and Permeabilization on the Mass Density of Single Cells: A Surface Plasmon
Resonance Imaging Study." *Frontiers in Chemistry* 7 (August): 588.
- Di Guardo, G. 2015. "Lipofuscin, Lipofuscin-like Pigments and Autofluorescence." *European*
*Journal of Histochemistry: EJH* 59 (1): 2485.
- Fischer, Andrew H., Kenneth A. Jacobson, Jack Rose, and Rolf Zeller. 2008. "Fixation and
Permeabilization of Cells and Tissues." *CSH Protocols* 2008 (May): db.top36.
- Fox, C. H., F. B. Johnson, J. Whiting, and P. P. Roller. 1985. "Formaldehyde Fixation." *The*
*Journal of Histochemistry and Cytochemistry: Official Journal of the Histochemistry Society*
33 (8): 845–53.
- Franek, Michal, Lenka Koptašíková, Jíří Mikšátko, David Liebl, Eliška Macíčková, Jakub
Pospíšil, Milan Esner, Martina Dvořáčková, and Jíří Fajkus. 2024. "In-Section Click-iT
Detection and Super-Resolution CLEM Analysis of Nucleolar Ultrastructure and Replication
in Plants." *Nature Communications* 15 (1): 2445.
- Fulton, Kara A., and Kevin L. Briggman. 2021. "Permeabilization-Free En Bloc
Immunohistochemistry for Correlative Microscopy." *eLife* 10 (May).
<https://doi.org/10.7554/eLife.63392>.
- Furuta, Takahiro, Kenta Yamauchi, Shinichiro Okamoto, Megumu Takahashi, Soichiro Kakuta,
Yoko Ishida, Aya Takenaka, et al. 2022. "Multi-Scale Light Microscopy/electron Microscopy
Neuronal Imaging from Brain to Synapse with a Tissue Clearing Method, ScaleSF."
*iScience* 25 (1): 103601.
- German, D. C., M. C. Ng, C. L. Liang, A. McMahon, and A. M. Iacopino. 1997. "Calbindin-D28k
in Nerve Cell Nuclei." *Neuroscience* 81 (3): 735–43.

Gong, Belvin, Karl D. Murray, and James S. Trimmer. 2016. "Developing High-Quality Mouse
Monoclonal Antibodies for Neuroscience Research - Approaches, Perspectives and
Opportunities." *New Biotechnology* 33 (5 Pt A): 551–64.

Huston, J. S., D. Levinson, M. Mudgett-Hunter, M. S. Tai, J. Novotný, M. N. Margolies, R. J.
Ridge, R. E. Brucoleri, E. Haber, and R. Crea. 1988. "Protein Engineering of Antibody
Binding Sites: Recovery of Specific Activity in an Anti-Digoxin Single-Chain Fv Analogue
Produced in *Escherichia Coli*." *Proceedings of the National Academy of Sciences of the
United States of America* 85 (16): 5879–83.

Ichikawa, Takehiko, Dong Wang, Keisuke Miyazawa, Kazuki Miyata, Masanobu Oshima, and
Takeshi Fukuma. 2022. "Chemical Fixation Creates Nanoscale Clusters on the Cell Surface
by Aggregating Membrane Proteins." *Communications Biology* 5 (1): 487.

Im, Sun-Woo, Hee Yong Chung, and Young-Ju Jang. 2017. "Development of Single-Chain Fv of
Antibody to DNA as Intracellular Delivery Vehicle." *Animal Cells and Systems* 21 (6): 382–
87.

Januszewski, Michał, Jörgen Kornfeld, Peter H. Li, Art Pope, Tim Blakely, Larry Lindsey,
Jeremy Maitin-Shepard, Mike Tyka, Winfried Denk, and Viren Jain. 2018. "High-Precision
Automated Reconstruction of Neurons with Flood-Filling Networks." *Nature Methods* 15 (8):
605–10.

Kiernan, John A. 2000. "Formaldehyde, Formalin, Paraformaldehyde And Glutaraldehyde: What
They Are And What They Do." *Microscopy Today* 8 (1): 8–13.

Kim, Eunhee G., Jieun Jeong, Junghyeon Lee, Hyeryeon Jung, Minho Kim, Yi Zhao, Eugene C.
Yi, and Kristine M. Kim. 2020. "Rapid Evaluation of Antibody Fragment Endocytosis for
Antibody Fragment-Drug Conjugates." *Biomolecules* 10 (6).
<https://doi.org/10.3390/biom10060955>.

Li, Tengfei, Matthias Vandesquille, Fani Koukoulis, Clémence Duffeffant, Ihsen Youssef, Pascal
Lenormand, Christelle Ganneau, et al. 2016. "Camelid Single-Domain Antibodies: A
Versatile Tool for in Vivo Imaging of Extracellular and Intracellular Brain Targets." *Journal of
Controlled Release: Official Journal of the Controlled Release Society* 243 (December): 1–
10.

Lu, Xiaotang, Xiaomeng Han, Yaron Meirovitch, Evelina Sjöstedt, Richard L. Schalek, and Jeff
676 W. Lichtman. 2023. "Preserving Extracellular Space for High-Quality Optical and
677 Ultrastructural Studies of Whole Mammalian Brains." *Cell Reports Methods* 3 (7): 100520.

Mahou, Pierre, Maxwell Zimmerley, Karine Loulier, Katherine S. Matho, Guillaume Labroille,
Xavier Morin, Willy Supatto, Jean Livet, Delphine Débarre, and Emmanuel Beaurepaire.
2012. "Multicolor Two-Photon Tissue Imaging by Wavelength Mixing." *Nature Methods* 9
(8): 815–18.

Marmorstein, Alan D., Lihua Y. Marmorstein, Hirokazu Sakaguchi, and Joe G. Hollyfield. 2002.
"Spectral Profiling of Autofluorescence Associated with Lipofuscin, Bruch's Membrane, and
Sub-RPE Deposits in Normal and AMD Eyes." *Investigative Ophthalmology & Visual
Science* 43 (7): 2435–41.

Mitchell, Keith G., Belvin Gong, Samuel S. Hunter, Diana Burkart-Waco, Clara E. Gavira-O'Neill,
Kayla M. Templeton, Madeline E. Goethel, et al. 2023. "High-Volume Hybridoma
Sequencing on the NeuroMabSeq Platform Enables Efficient Generation of Recombinant
Monoclonal Antibodies and scFvs for Neuroscience Research." *Scientific Reports* 13 (1):
16200.

Monnier, Philippe P., Robin J. Vigouroux, and Nardos G. Tassew. 2013. "In Vivo Applications of
Single Chain Fv (Variable Domain) (scFv) Fragments." *Antibodies* 2 (2): 193–208.

Pudavar, Haridas, Judith Reddington, Jason A. Junge, Scott E. Fraser, and Giulia Ossato.
2024. "STELLARIS 8 DIVE: A Rainbow of Possibilities with Multiphoton Excitation and
Lifetime-Based Information." In *Multiphoton Microscopy in the Biomedical Sciences XXIV*,
12847:32–33. SPIE.

- Schmid, Benjamin, Gopi Shah, Nico Scherf, Michael Weber, Konstantin Thierbach, Citlali Pérez
Campos, Ingo Roeder, Pia Aanstad, and Jan Huisken. 2013. "High-Speed Panoramic Light-
Sheet Microscopy Reveals Global Endodermal Cell Dynamics." *Nature Communications* 4:
2207.
- Schmidt, Hartmut, Oliver Arendt, Edward B. Brown, Beat Schwaller, and Jens Eilers. 2007.
"Parvalbumin Is Freely Mobile in Axons, Somata and Nuclei of Cerebellar Purkinje
Neurons." *Journal of Neurochemistry* 100 (3): 727–35.
- Shapson-Coe, Alexander, Michał Januszewski, Daniel R. Berger, Art Pope, Yuelong Wu, Tim
Blakely, Richard L. Schalek, et al. 2021. "A Connectomic Study of a Petascale Fragment of
Human Cerebral Cortex." *bioRxiv*. <https://doi.org/10.1101/2021.05.29.446289>.
- Thavarajah, Rooban, Vidya Kazhiyur Mudimbaimannar, Joshua Elizabeth, Umadevi
Krishnamohan Rao, and Kannan Ranganathan. 2012. "Chemical and Physical Basics of
Routine Formaldehyde Fixation." *Journal of Oral and Maxillofacial Pathology: JOMFP* 16
(3): 400–405.
- Thiel, M. A., D. J. Coster, S. D. Standfield, H. M. Brereton, C. Mavrangelos, H. Zola, S. Taylor,
712 A. Yusim, and K. A. Williams. 2002. "Penetration of Engineered Antibody Fragments into
713 the Eye." *Clinical and Experimental Immunology* 128 (1): 67–74.
- Tsuruel, Shlomo, Sagi Gudes, Ryan W. Draft, Alexander M. Binshtok, and Jeff W. Lichtman.
2015. "Multispectral Labeling Technique to Map Many Neighboring Axonal Projections in
the Same Tissue." *Nature Methods* 12 (6): 547–52.
- Weisser, Nina E., and J. Christopher Hall. 2009. "Applications of Single-Chain Variable
Fragment Antibodies in Therapeutics and Diagnostics." *Biotechnology Advances* 27 (4):
502–20.
- Wittrup, Anders, Si-He Zhang, Gerdy B. ten Dam, Toin H. van Kuppevelt, Per Bengtson, Maria
Johansson, Johanna Welch, Matthias Mörgelin, and Mattias Belting. 2009. "ScFv Antibody-
Induced Translocation of Cell-Surface Heparan Sulfate Proteoglycan to Endocytic
Vesicles." *The Journal of Biological Chemistry* 284 (47): 32959–67.

REVIEWERS' COMMENTS

Reviewer #1 (Remarks to the Author):

The authors have developed a method employing fluorescent single-chain variable fragments (scFvs) for conducting multiplexed detergent-free immunolabeling and volumetric-correlated-light-and-electron-microscopy (vCLEM) on the same samples. In this manuscript, they detail the development of eight fluorescent scFvs targeting specific markers crucial for brain studies. Through experimentation, six fluorescent probes were successfully visualized in the cerebellum using confocal microscopy with spectral unmixing, followed by vEM analysis of the identical sample. The outcomes reveal an exceptional blend of ultrastructure alongside multiple fluorescence channels, offering valuable insights into cellular composition and organization.

This approach facilitated the documentation of a previously poorly characterized cell type and the precise subcellular localization. Leveraging scFvs derived from existing monoclonal antibodies opens up the possibility of generating numerous such probes, which could significantly enhance molecular overlays for connectomic investigations.

This study represents a significant advancement in the field of vCLEM and holds great promise for researchers seeking to unravel intricate neuronal networks. The revised version of the manuscript effectively addresses key questions and concerns. I don't have further comments.

Reviewer #2 (Remarks to the Author):

The revised manuscript thoroughly addresses all of the concerns raised in my initial review. I think the manuscript will have broad appeal and I appreciate the effort put in to address the referees' comments. I strongly recommend publication in Nature Communications.

**REVIEWER COMMENTS**

**Reviewer #1 (Remarks to the Author):**

*The authors have developed a method employing fluorescent single-chain variable fragments*
*(scFvs) for conducting multiplexed detergent-free immunolabeling and volumetric-correlated-*
*light-and-electron-microscopy (vCLEM) on the same samples. In this manuscript, they detail the*
*development of eight fluorescent scFvs targeting specific markers crucial for brain studies.*
*Through experimentation, six fluorescent probes were successfully visualized in the cerebellum*
*using confocal microscopy with spectral unmixing, followed by vEM analysis of the identical*
*sample. The outcomes reveal an exceptional blend of ultrastructure alongside multiple*
*fluorescence channels, offering valuable insights into cellular composition and organization.*

*This approach facilitated the documentation of a previously poorly characterized cell type and*
*the precise subcellular localization. Leveraging scFvs derived from existing monoclonal*
*antibodies opens up the possibility of generating numerous such probes, which could*
*significantly enhance molecular overlays for connectomic investigations.*

*This study represents a significant advancement in the field of vCLEM and holds great promise*
*for researchers seeking to unravel intricate neuronal networks. The revised version of the*
*manuscript effectively addresses key questions and concerns. I don't have further comments.*

We thank the reviewer's comments. There's nothing to be further addressed.

**Reviewer #2 (Remarks to the Author):**

*The revised manuscript thoroughly addresses all of the concerns raised in my initial review. I*
*think the manuscript will have broad appeal and I appreciate the effort put in to address the*
*referees' comments. I strongly recommend publication in Nature Communications.*

We thank the reviewer's comments. There's nothing to be further addressed.
